

# Assessing the impact of rainfall seasonality anomalies
# on catchment-scale water balance components
Paolo Nasta[1,*], Carolina Allocca[1], Roberto Deidda[2], Nunzio Romano[1,3]
[1] Department of Agricultural Sciences, AFBE Division, University of Napoli Federico II, Portici (Napoli), Italy.
[2] Department of Civil and Environmental Engineering and Architecture, University of Cagliari, Cagliari, Italy.
[3] The Interdepartmental Research Center for Environment (C.I.R.AM.), University of Napoli Federico II, Napoli, Italy.
*Correspondence to*: Paolo Nasta (paolo.nasta@unina.it)
**Keywords:** Mediterranean climate, Budyko curve, drought, Standardized Precipitation Index, SWAT model, Upper
Alento River Catchment
**Abstract.** Water balance components at catchment scale are strongly related to annual rainfall amount. Nonetheless,
water resources availability in Mediterranean catchments depends also on rainfall seasonality. Indeed, a high percentage
of annual rainfall occurs between late fall and early spring and feeds natural and artificial water reservoirs. This amount
of water stored in the mild-rainy season is used to offset rainfall shortages in the hot-dry season (between late spring
and early fall). Observed seasonal anomalies in historical records are quite episodic, but an increase of their frequency
might exacerbate water stress or water excess if the rainy season shortens or extends its duration, e.g. due to climate
change. Hydrological models are useful tools to assess the impact of seasonal anomalies on the water balance
components and this study evaluates the sensitivity of water yield, evapotranspiration and groundwater recharge on
changes in rainfall seasonality by using the Soil Water Assessment Tool (SWAT) model. The study area is the Upper
Alento River Catchment (UARC) in southern Italy where a long time-series of daily rainfall is available from 1920 to
2018. To assess seasonality anomalies, we compare two distinct approaches: a "static" approach based on the
Standardized Precipitation Index (SPI), and a "dynamic" approach that identifies the rainy season by considering
rainfall magnitude, timing, and duration. The former approach rigidly selects three seasonal features, namely rainy, dry,
and transition seasons, the latter being occasionally characterized by similar properties to the rainy or dry periods. The
"dynamic" approach, instead, is based on a time-variant duration of the rainy season and enables to corroborate the
aforementioned results within a probabilistic framework. A dry seasonal anomaly is characterized by a decrease of 241
mm in annual average rainfall inducing a concurrent decrease of 116 mm in annual average water yield, 60 mm in
actual evapotranspiration and 66 mm in groundwater recharge. We show that the Budyko curve is sensitive to the
seasonality regime in UARC by questioning the implicit assumption of temporal steady-state between annual average
dryness and evaporative index. Although the duration of the rainy season does not exert a major control on water



balance, we have been able to identify seasonal-dependent regression equations linking water yield to dryness index
over the rainy season.

## 1. Introduction

The rainfall regime of the Mediterranean climate is characterized by the alternation of wet and dry periods within the
year, with an evident out-of-phase seasonal behavior of precipitation and temperature patterns. Indeed, the majority of
the annual amount of rainfall is concentrated in the late fall and winter months, while summer is usually hot and quite
dry. Rainfall seasonality plays a fundamental role in planning and managing water resources in countries subject to a
Mediterranean climate.
Scarce rainfall supply, combined with high evapotranspiration losses and excessive consumption of water (agricultural,
industrial, and recreational uses, hydroelectric power generation, as well as civil uses being often increased by the
tourism pressure) induces water stress during summer. Therefore, it is necessary to store water during the rainy period
to cope with the "uncertain" duration of adverse water deficit conditions during the dry period. Supply-water
infrastructures necessitate high investment costs that strongly depend on the expected balance between the amount of
water supplied in the rainy period and the amount of water lost and consumed during the dry season. The amount of
rainfall in each season can be suitably decomposed and simulated considering the following three main components: *i)*
duration of the seasons; *ii)* occurrence probability of a daily rainfall event in each season; *iii)* mean depth of daily
rainfall events in each season (Van Loon et al., 2014). A combination of the last two factors determines the rainfall
magnitude in each season (Feng et al., 2013).
A very low or very high amount of water (exceeding a certain threshold value for a specified return period and duration)
that is supplied during the rainy period can be interpreted as a seasonal precipitation anomaly and is usually observed
episodically in a historical multi-decadal time-series of annual rainfall values. The seasonal precipitation anomalies



depend mainly on a combination of the duration of the wet season and its rainfall magnitude. These two factors should
be taken in due account when planning supply-water infrastructures (Apurv et al., 2017). The most recent reports
released by the Intergovernmental Panel on Climate Change (IPCC) warn on projected increase in seasonal anomalies
induced by global warming in the Mediterranean region, with a remarkable decrease in annual precipitation and
warming-enhanced evapotranspiration associated with rather severe and prolonged droughts, as recently observed in
southern Europe in 2003, 2015, and 2017 (Mariotti et al., 2008; Laaha et al., 2017; Hanel et al., 2018).
Studies under way in the Upper Alento River Catchment (UARC) offer a good chance to understand the effects of
rainfall seasonal uncertainty on water supply generation given the presence of a multi-purpose earthen dam constructed
to regulate water for irrigation, hydro-power generation, flood control, and drinking purposes. The main research
question, also solicited or prioritized somehow by local stakeholders in their decision-making processes, can be
expressed as follows: "*What is the impact of rainfall seasonality anomalies on annual-average (or seasonal-average)*
*water supply and what happens if the Alento River catchment (ARC) will experience several consecutive years of lower-*
*than-expected rainfall events?*"
To deal with at least the first part of the above research question, a prime objective is the quantification of the effects
exerted by rainfall seasonality on water balance components. With a view to positive interactions with stakeholders,
end-users, and professionals, we performed this task by implementing the well-known and well-validated Soil Water
Assessment Tool (SWAT) model whereas a particular attention is devoted to the computation of water yield supplying
the artificial reservoir bounded by the "Piano della Rocca" earthen dam in ARC (Romano et al., 2018).
Many authors attempted to quantify the rainfall seasonality by using different approaches (Ayoade, 1970; Markham
1970; Nieuwolt, 1974; Oliver, 1980; Walsh and Lawler, 1981; Zhang and Qian, 2003; Martin-Vide, 2004; Potter et al.,
2005; Feng et al., 2013; de Lavenne and Andréassian, 2018). The Precipitation Concentration Index (PCI) proposed by
Oliver (1980) is the most popular approach for quantifying the year-round precipitation distribution in a given study



area (Raziei, 2018). Sumner et al. (2001) analyzed the spatial and temporal variation of precipitation seasonality over
the eastern and southern Spain by using the seasonality index (SI). The SI indicator was also utilized for examining the
spatial and temporal variability of precipitation seasonality in Greece (Livada and Asimakopoulos 2005), USA (Pryor
and Schoof 2008) and northern Bangladesh (Bari et al. 2016). Under the typical Mediterranean climate of Sardinia
(Italy), Corona et al. (2018) used the SI indicator to evaluate the role of precipitation seasonality on runoff generation.
The goal of this study is to characterize the rainfall seasonality and its anomalies by using two approaches. A first
approach, which is hereafter referred to as the static approach, is based on the analysis of the Standardized Precipitation
Index (SPI). The second approach, instead, exploits the seasonality characterization proposed by Feng et al. (2013) and
can be viewed as a dynamic approach. As far as we are aware, there is still a lack of knowledge about the effects of
possible changes in rainfall seasonality on the water balance of a catchment subject to a Mediterranean climate, and the
analyses presented in this paper aims primarily at contributing to fill this gap.
**2. Study area and experimental analyses**
The Upper Alento River Catchment (UARC) is situated in the Southern Apennines (Province of Salerno, Campania,
southern Italy) and has a total drainage area of about 102 km$^2$. The "Piano della Rocca" dam is an earthen embankment
with impervious core that has been operating since 1995. The area consists mostly of relatively poor-permeable
arenaceous-clayey deposits and secondarily of arenaceous-marly-clayey and calcareous-clayey deposits (Romano et al.,

92   2018).

A weather station managed by the Italian Hydrological Service is located in the village of Gioi Cilento and provides a
dataset of daily rainfall values covering the period 1920-2018 (about 90 years), with an interruption of 9 years (1942-
1950) that straddled World War II (Nasta et al., 2017). The total (cumulative) annual depth of precipitation derived
from the daily rainfall time series of the entire available period is characterized by a mean of 1,229.3 mm, a median
value of 1,198.3 mm, a standard deviation (Std. Dev.) equal to 295.9 mm, and a coefficient of variation (CV) equal to





24.1%; the mean and median values are quite close indicating that this available dataset follows a normal distribution
closely. The variability exhibited by the monthly time series of rainfall depths is instead summarized in Table 1 and
Figure 1. A large amount of precipitation occurs in the months from October to March, a period commonly identified as
a wet period of a hydrological year, and accounts for about 68% of the mean annual rainfall (i.e. 834.9 mm over 1,229.3
mm) (see Table 1 and Figure 1). November is the wettest month with an average monthly rainfall depth of 152.2 mm
(about 14% of mean annual rainfall). In contrast, lower means of monthly rainfall depth are concentrated from April to
September, which is commonly identified as a dry period of a hydrological year, with a cumulative rainfall depth over
this period of 343.7 mm with respect to mean yearly value of 1,229.3 mm, and hence representing about 31% of the
mean annual rainfall. July is the driest month with a mean monthly rainfall depth of 17.6 mm (i.e. 1.6% of the yearly
rainfall depth).
*Please insert Fig. 1 here*
*Please insert Table 1 here*
Within the monitoring activities of the MOSAICUS project (Nasta et al., 2013; Romano et al., 2018), an automated
weather station was installed in 2004 close to the village of Monteforte Cilento and equipped with sensors for
precipitation, wind speed and direction, air temperature and relative humidity, and solar radiation, to record these
meteorological variables at 15 min intervals. The data set of daily rainfall values (1920-2018) recorded at the weather
station of Gioi Cilento will be used to assess rainfall seasonality. The statistical distributions of weather data recorded at
the weather station of Monteforte Cilento (2004-2018) will be used to calculate potential evapotranspiration as
described in Section 3.
In this study we used the most recent available land-use map drawn on 2015 by using second-level CORINE
(Coordination of Information on the Environment) Land-Cover classes (CORINE 2006 land cover dataset;
http://www.eea.europa.eu): forest, arable land (annual crops), permanent crops (orchards, vineyards, olive groves and
fruit trees), pasture, urban fabric, and water bodies. Forest (evergreen and deciduous trees, and multi-stem evergreen



sclerophyllous Mediterranean shrubs) and agricultural (arable land, permanent crops and orchards) cover about 70%
and 20% of the catchment (Nasta et al., 2017).
**3. Parameterization of the SWAT Model**
The Soil Water Assessment Tool (SWAT) is a bucket-type, semi-distributed hydrological model operating on a daily
time scale and at a catchment spatial scale (Arnold et al., 1998). The main components of the water balance equation are
the daily change in water storage ($\Delta WS$) as affected by rainfall ($R$), actual evapotranspiration ($ET_a$), groundwater
recharge ($GR$), and water yield ($WY$). Water yield is given by the contribution of surface runoff, groundwater
circulation, and lateral flow within the soil profile, and is partially depleted by transmission losses from tributary
channels and water abstractions. All variables are expressed in units of mm of water height.
The boundary forcings are rainfall ($R$) and potential evapotranspiration ($ET_p$) computed on a daily basis. SWAT is
based on the concept of Hydrological Response Units (HRUs), which are areas identified by similarities in soil, land
cover, and topographic features. A 5-m Digital Elevation Model (DEM) of the study area was used to determine the
catchment boundaries, the hydrographic network, and thirteen distinct HRUs. Catchment-lumped parameters are
assigned to each HRU through look-up tables. Known parameters were assigned according to model set up presented in
Nasta et al. (2017). Nine parameters were calibrated to achieve the best model fit between simulated and measured
monthly water yield data recorded from 1995 and 2004 (Nasta et al., 2017). Such hydrological parameters include the
soil evaporation and compensation factor, plant uptake compensation factor, Manning's value for overland flow,
baseflow recession constant (groundwater flow response to changes in recharge), groundwater delay time, groundwater
"revap" coefficient (controlling water that moves from the shallow aquifer into the unsaturated zone), Manning's
coefficient for the main channel, effective hydraulic condition in the main channel alluvium, and bank storage recession
curve. Model performance proved to be satisfactory at monthly time scale.



This study is based on modelling scenarios implemented in SWAT through a Monte Carlo approach, where each
simulation is 3-year long. Results from the first 2-year warm-up period are discarded, while water balance components
simulated for the third year are stored for subsequent analysis. Initial soil water storage is set as field capacity. The
rainfall data will be generated for the static and dynamic approaches (described below) using a probability setting
calibrated on daily rainfall values recorded at the Gioi Cilento weather station (1920-2018). The meteorological data
recorded at the second automated weather station (close to the village of Monteforte Cilento) will be used for statistical
analysis at monthly time scale: results will be provided as input to SWAT in order to randomly generate daily reference
evapotranspiration by using the Penman-Monteith equation (Allen et al., 1998).
**4. Determination of rainfall seasonality**
**4.1. Static approach based on the SPI drought index**
The intra-annual rainfall regime under Mediterranean climate can be characterized through the partitions of annual
rainfall depth among different seasons (Paz and Kutiel, 2003; Kutiel and Trigo, 2013). The seasonal pattern occurring in
the study area is based on long-term monthly rainfall time series through the Standardized Precipitation Index (SPI).
SPI is a probability index developed to classify rainfall anomalies and often employed as an indicator of potential
(meteorological) droughts over many time scales (McKee et al., 1993; Hayes et al., 1999). The computation of SPI
should rely on long-term rainfall datasets (e.g. 30 years, according to climatological standards), and is usually obtained
by projecting a Gamma distribution fitted on rainfall depths cumulated on 3, 6, 12, 18, or 24 months (referred to as SPI-
3, SPI-6, SPI-12, SPI-18, or SPI-24, respectively) into a standardized normal distribution. Short-term SPI (e.g. 3-month
time scale) can provide useful information for crop production and soil moisture supply, while long-term SPI (e.g. 12-
or 24-month time scale) can give insights on water availability for groundwater recharge. Negative SPI-values indicate
lower-than-expected rainfall, whereas positive SPI-values refer to wetter-than-expected months. To quantify the degree
of departure from median conditions, McKee et al. (1993) proposed a rainfall regime classification. Since SPI is given
in units of standard deviation from the standardized mean, this statistical index enables also the precipitation anomaly to



be identified through the magnitude of its value: values ranging from -0.99 to +0.99 are considered near normal, from
+1.00 to +1.49 (or from -1.49 to -1.00) indicates moderately wet (or moderately dry) periods, from +1.50 to +1.99 (or
from -1.99 to -1.50) very wet (or very dry) periods, and above +2.00 (or below -2.00) extremely wet (or extremely dry)
periods. Therefore, the extent of SPI departure from the mean (i.e. from the zero value) gives a probabilistic measure of
the severity of a wet (if positive) or dry (if negative) period. By exploiting the properties of the (standard) normal
distribution, the probabilities to obtain SPI-values greater than +1, +2, and +3 (or lower than -1, -2, and -3) are 15.9%,
2.28% and 0.135%, respectively.
In order to emphasize the seasonal cycle of intra-annual rainfall patterns within a probabilistic framework, we slightly
modified the common SPI application by fitting the Gamma distribution on all monthly rainfall depths, i.e. pooling
together observations from all months in each year. In such a way, the months characterized by SPI-values below,
around or above the zero line can be assumed to belong to the dry, transition or wet seasons, respectively.
**4.2. Dynamic approach based on duration of the wet season proposed by Feng et al. (2013)**
According to Feng et al. (2013), the Dimensionless Seasonality Index (DSI) is based on the concept of relative entropy
and quantifies the rainfall concentration occurring in the wet season. DSI is zero when the average annual rainfall is
uniformly distributed throughout the year and maximized at 3.585 when maximum average annual rainfall is
concentrated in one single month (Pascale et al., 2016); see Appendix for details. Feng et al. (2013) proposed to
describe the rainfall seasonality through the following three components: annual rainfall depth (magnitude), centroid
(timing), and spread (duration) of the wet season (see also Pascale et al., 2015; Sahani et al., 2018). Following this
framework, the hydrological year is assumed to start from the driest month and proceeds for the subsequent 12 months,
rather than starting at a prescribed month (e.g. on April, according to a conventional way). Specifically, we assumed
that the duration of the wet season follows a normal distribution, with mean and standard deviation estimated from the
90 durations obtained for each year by applying to the Gioi Cilento time series the procedure proposed by Feng et al.
(2013) and briefly resumed in the Appendix.



### 4.3 Set up of Monte-Carlo rainfall scenarios in SWAT


Rainfall seasonality anomalies, although episodic, can affect the water balance components at catchment scale. As
suggested by Domínguez-Castro et al. (2019), the impact of such anomalies can be quantified within a probabilistic
framework. For the Upper Alento River Catchment (UARC), we evaluated the effects of seasonal anomalies by running
SWAT simulations with synthetic rainfall time series considering different hypotheses (scenarios) of alternations of
seasons, according to the "static" and the "dynamic" approaches described above. In each season, we assumed that
rainfall evolution in time can be represented by a stochastic Poisson point process of daily rainfall occurrences, with
daily rainfall depth following a proper probability distribution. Synthetic rainfall time series were then generated
keeping constant parameters of the Poisson process and daily rainfall parent distribution in each season.
A preliminary analysis was conducted to investigate the best parent distribution for observed rainfall daily depths. With
this aim, we used the L-moment ratios diagram proposed by Hosking (1990) (see also Vogel and Fennessey, 1993) as
diagnostic tool. Results are shown in Figure 2 where the L-skewness and L-kurtosis computed on the time series left-
censored with a threshold of 3 mm (large filled circle) is compared with the theoretical expectation of the same L-
moment ratios for several probability distributions commonly adopted in statistical hydrology. It is apparent that ideal
candidate as parent distribution is the Generalized Pareto distribution (GPd), although it is also worthwhile noticing that
sample estimation of L-skewness and L-kurtosis (0.3437, 0.1706) is very close to the expected values for an exponential
distribution (1/3,1/6). As a visual support of this preliminary analysis, the exponential probability plot in Figure 3
compares the empirical cumulative distribution function F(x) of the observed time series (circles) with the fitted GPd
(dashed line) and the fitted exponential distribution (continuous line). It is apparent that the two models are very close
each other for the whole body of observation, with only a slight departure of the GPd from the straight line charactering
the exponential distribution due to a very light right tail. These evidences made us confident in adopting the single-
parameter exponential model as parent distribution for series partitioned according to the seasons defined above,
reducing in such a way the uncertainty related to the additional shape parameter of the GPd. Finally, it is worthwhile
mentioning that both distributions shown in Figure 3 were fitted applying the Multiple-Threshold-Method (MTM) by



Deidda (2010) on a range of thresholds from 2.5 to 12.5 mm to prevent biases due to very low records and data
discretization (Deidda, 2007). The MTM was then applied to estimate the exponential parameter $\eta$ (mm) and the
probability occurrence of rainy days $\lambda$ ($d^{-1}$) for each considered season.
For each scenario pertaining to either the "static" or "dynamic" approach, we generated 10,000 equi-probable
realizations of synthetic daily rainfall time series, each 3-year long, according to a stochastic Poisson point process
model. In each modelling scenario, the synthetic time series was then used as input of the SWAT model to evaluate the
effects on the water balance components in UARC. The first two years represent warm-up simulations and thus
discarded, while only results for the third year were stored for subsequent analyses presented in the next section. For the
former approach the alternation of seasons was fixed, as already pointed out, while for the "dynamic" approach the
duration of wet season in each year was randomly drawn from a normal distribution (with mean equal to 2.71 months
and standard deviation equal to 0.28 months, estimated from the Gioi Cilento daily rainfall dataset).
*Please insert Fig. 2 here*
*Please insert Fig. 3 here*

**5. Results and discussion**
**5.1. Static approach**
Observed temporal evolution of SPI-6 in our time series (see grey bars in Fig. 4) highlights prolonged droughts in
between the 1980s and the 1990s and prolonged wet periods in the last decade when SPI-6 values above the threshold
+2 occurred in 2008, 2010, and 2012. Yet, by splitting the frequency distribution of the SPI-6 values in two sub-groups,
one in the first 45 years and a second one in the last 45 years, we observe a general drying trend. In the first sub-group
the probabilities to obtain SPI-6>+1 and SPI-6<-1 are 17.9% and 7.6%, respectively. In contrast, in the second sub-
group there is a general increase of negative SPI-6 values by turning the probability into 11.9% to obtain SPI-6>+1 and
19.3% to obtain get SPI-6<-1. By analyzing daily rainfall datasets recorded at 55 weather stations located in the



Basilicata Region nearby UARC (characterized by similar climatic conditions), Piccarreta et al. (2013) observed a
general decreasing trend in the mean annual rainfall over the period 1951–2010 mainly due to the autumn-winter
decrease of precipitation.
*Please insert Fig. 4 here*
We discuss now about the results pertaining to the calculation of the seasonal SPI-values. Rainfall seasonality under a
Mediterranean climate can be assumed to be roughly represented by the alternation of two 6-month seasons,
characterized by positive and negative SPI-values (wet and dry season, respectively) (Rivoire et al., 2019). The
temporal evolution of the SPI-values is represented by the grey bars in Fig. 5a and highlights the seasonal cycle within
each year, whereas their 12-month moving average (magenta line in Fig. 5a) oscillates around the zero-value with
prolonged dry periods in between the 1980s and the 1990s and prolonged wet periods between the 2000s and the 2010s.
Fig. 5b shows the box and whiskers plots of the SPI-values for each month of the year, thus depicting the monthly
distribution of this index throughout the available recorded period. The median SPI-values (central red line in the blue
boxes) are negative only from May to August and positive from September to April, even though the whiskers
(identified by the two lines at the 25th and 75th percentile) denote the presence of a relatively large variability in almost
all months. A closer inspection of this graph enables one to identify three main seasonal features: *i*) a dry period from
May till August with median values below zero; *ii*) a rainy period from November till February with median values
above zero; *iii*) two transition periods from wet to dry (March and April) and from dry to wet (September and October)
with median values near zero. We are aware that the median values in March, April and October of the transition season
are above zero, rather than "near" zero, but we remind that the Mediterranean climate in UARC is sub-humid mainly
due to orographic influences. However, this approach can be considered "static" since the subdivision of the twelve
months in three groups is rigid even though months in the transition periods are characterized by the highest SPI-values
variability. This outcome refines the initial working hypothesis of seasonal alternation of two semesters with random
durations.





*Please insert Fig. 5 here*
The frequency distributions of the SPI-values computed over the rainy, dry, and transition seasons are illustrated in
Fig.5c-5d-5e. The wet season (depicted by the blue histograms) is characterized by probabilities to have SPI-values
greater than 0, +1, +2, and +3 of 80.6%, 30.5%, 1.9%, and 0.3%, respectively. The dry season (depicted by the red
histograms) is associated with SPI-values lower than 0, -1, -2, and -3 with probabilities of 78.1%, 31.1%, 0.56% and
0.1%, respectively. Conversely, we warn that probabilities to have positive SPI-values in the transition season are of
63.3% instead of the expected 50% if the hypothesis was "perfectly true". We therefore considered three scenarios, each
with fixed and recurrent alternation of seasons during the hydrological year: *i*) a "reference scenario" with a 4-month
wet season (NDJF), a 4-month dry season (MJJA), and a 4-month transition season (MA from wet to dry and SO from
dry to wet); *ii*) a "dry scenario", which mimics an extreme drought anomaly, characterized by a prolonged 8-month dry
season (from March to October) and abrupt alternations with the 4-month wet season (NDJF), without any transition
season; *iii*) a "wet scenario", which mimics an extreme rainy anomaly, characterized by a prolonged 8-month wet
season (from September to April) and abrupt alternations with the 4-month dry season (MJJA), again with no transition
season.
In light of the aforementioned results, the two Poisson parameters ($\eta$ and $\lambda$) describing daily rainfall values were
calculated for each of the three seasons in the "reference scenario" and they are then also used for developing synthetic
simulations of rainfall time series in the "dry" and "wet" scenarios (see Table 2).
*Please insert Table 2 here*

**5.2. Dynamic approach**
The centroid of the monthly rainfall distribution measured at the Gioi Cilento weather station (in the 90 years between
1920 and 2018) indicates that the wet season is centered in the second half of December, while its average duration is
about 5.44 months (see Fig. 6). Nonetheless, it is worth noting the occurrence of a few extreme situations: the severe
drought spell recorded in 1985 caused a minimum duration of about 4 months of the rainy period, while the year 1964





registered a maximum duration of about 7.0 months. The term "dynamic" for this approach stems mainly from the fact

that the duration of the rainy period is time-variant throughout the years.

*Please insert Fig. 6 here*

The Mann-Kendall nonparametric test (Mann, 1945; Kendall, 1975) is used to evaluate possible decreasing, increasing,

or absence of temporal trends on the DSI (Feng et al., 2013) or the seasonality index (SI) proposed by Walsh and

Lawler (1981). This test did not highlight significant trend on DSI and SI at 0.05 significance level ($z_c$-values of -0.0027

and 0.0030, respectively). The stationarity in time of DSI (red line) and SI (green line) is also apparent from a perusal of

Fig. 7, where the linear regressions (dashed and dotted for DSI and SI, respectively) are characterized by very weak

downward slopes.

*Please insert Fig. 7 here*

Under the "dynamic" approach, we consider the alternation of only two seasons (wet and dry) with random durations of

the rainy period. Figure 8a shows the time series of the estimated duration of the wet season in each year, while the

Lilliefors statistical test has verified at 5% significance level that observed data (Fig. 8b) belongs to a normal

distribution (Lilliefors, 1967). The dry seasons were consequently obtained as the complement to the wet seasons. In

this case, the two Poisson parameters ($\eta$ and $\lambda$) for modeling daily rainfall values were computed for the wet and dry

seasons (Table 3).

*Please insert Fig. 8 here*

*Please insert Table 3 here*

**5.3. Effects of rainfall seasonality anomalies on water balance by using the static approach**



The results obtained from the three scenarios pertaining to the "static" approach are presented using the descriptive
statistics of the water balance components at the annual time scale obtained from 10,000 SWAT simulation runs (Table
4). Reference scenario represents the normal situation with three seasons (dry, transition, and wet). Even though the
range of annual rainfall values is relatively large, the coefficient of variation (CV) is only 14%, implying that very low
and very high (outliers) annual rainfall depths occur occasionally. The water balance components, namely water yield
($WY$), actual evapotranspiration ($ET_a$), and groundwater recharge ($GR$), represent averagely 35%, 49%, and 16% of the
annual mean rainfall depth ($R$=1,229 mm). The annual rainfall depths for the other two scenarios (only two seasons
without the transition season) shift down to 988 mm (dry scenario) and up to 1,393 mm (wet scenario) and consequently
affect the water balance. When the dry season lasts 8 months (dry scenario), water yield, actual evapotranspiration, and
groundwater recharge decrease by 116 mm, 60 mm, and 66 mm, respectively, when compared to the reference scenario.
*Please insert Table 4 here*

In contrast, when the wet season lasts 8 months (wet scenario), the water yield, actual evapotranspiration, and
groundwater recharge increase by 93 mm, 21 mm, and 54 mm, respectively, when compared to the reference scenario.
Water yield, actual evapotranspiration, and groundwater recharge represent averagely 32%, 55%, and 13% of the annual
rainfall depth in the extreme dry season (dry scenario) and 38%, 45%, and 18% of annual rainfall depth in the extreme
wet season (wet scenario).
The decomposition of the annual results into the seasonal components highlight other interesting features that are
showed in Fig. 9 (boundary forcings) and in Fig. 10 (main water balance components). For the reference scenario the
seasonal rainfall depth is 201 mm, 436 mm, and 593 mm for the dry, transition, and wet seasons, respectively,
representing 16%, 35%, and 48% of the total annual rainfall (see Fig. 9a). Water yield depths span from 44 mm during
the dry season to 251 mm during the rainy season (see Fig. 10a). Almost 60% of annual water yield occurs over the wet
season, about 30% in the transition season, and about 10% in the dry season. In contrast, the actual evapotranspiration





depths are higher than rainfall depths in the dry season (269 mm) and lower than rainfall depths during the transition
(226 mm) and rainy (110 mm) seasons (see Fig. 10a).
*Please insert Fig. 9 here*
*Please insert Fig. 10 here*

Over the dry scenario (see Fig. 9b and 10b), the months belonging to the transition season become drier. The total
rainfall depths over the dry and wet seasons are 397 mm and 590 mm, respectively, whereas the extreme drought
anomaly causes precipitation loss only in the dry season with a consistent decrease of 239 mm of rainfall depth (Fig.
9b). The consequences of this situation on the average water balance components in the prolonged dry season lead to
significant deficits (Fig. 10b). Water yield loss in the dry season is 93 mm which represents 50% of water yield
obtained in the dry and transition seasons in reference scenario. The wet season (from November to February) provides
about 590 mm of water yield per year. The water lost by actual evapotranspiration is limited and represents only 10% of
$ET_a$ obtained in the dry and transition seasons in reference scenario (Fig. 10b).
In the wet scenario (see Fig. 9c and Fig. 10c), the months belonging to transition season turn wet (8 wet months and 4
dry months). Total rainfall depths in the dry and wet seasons are 200 mm and 1,193 mm (Fig. 9c). Rainfall depth
increases by 164 mm in the wet season (+14% than the one obtained in the wet and transition seasons in reference
scenario). Water yield gain in the wet season is 89 mm which represents 20% of water yield obtained in the wet and
transition seasons in reference scenario (Fig. 10c). The water lost by actual evapotranspiration is negligible.
**5.4. Effects of rainfall seasonality anomalies on water balance by using the dynamic approach**
The second approach for assessing the effect of rainfall seasonality extremes on water balance components is based on
the stochastic generation of the wet season durations from their normal distribution (see Fig. 8b). This approach helps
classify the results within a probabilistic framework according to the following duration classes: 3-4 months, 4-5
months, 5-6 months, 6-7 months, 7-8 months. Seasonal extremes (3-4 months and 7-8 months) have very low





occurrence probabilities (0.6% and 0.3%). Nonetheless it is interesting to analyze the effect of rainfall variability on
water yield (*WY*), actual evapotranspiration (*ET$_a$*) and groundwater recharge (*GR*). The most probable (62%) situation
occurs when the rainy period lasts 5-6 months. Under these circumstances, the mean annual rainfall depth is 1,275 mm,
whereas *WY*, *ET$_a$*, and *GR* represent 35%, 49%, and 16% of annual average rainfall depth, respectively. These
percentages are the same observed in reference scenario of the static approach. If the wet season shortens by one month
(23% probability), the mean annual rainfall depth decreases by 62 mm, whereas water yield depth by 33 mm (-7%). In
contrast, if the wet season is made up of 6-7 months (14% probability), the mean annual rainfall depth increases by 51
mm and water yield by 27 mm (+6%).
Extreme dry and extreme wet situations reflect similar results obtained from the dry and wet scenarios presented above.
A prolonged drought spell (i.e. lasting 3-4 months) leads to average rainfall loss of 130 mm per year inducing a
consistent annual decrease in both water yield (-68 mm) and groundwater recharge (-30 mm).  A prolonged wet season
(i.e. lasting 7-8 months), instead, causes an average rainfall to gain approximately 108 mm per year, hence yielding
annual increases in both water yield (+59 mm) and groundwater recharge (+12  mm). It is worth noting that the duration
of the rainy period does not seem to exert a major control on the water balance. The Pearson's linear correlation
coefficients between duration and average annual rainfall, water yield, and actual evapotranspiration are 0.22, 0.20, and
0.11, respectively.
*Please insert Table 5 here*

To further evaluate the hydrologic behavior of the study catchment, an issue deserving to be addressed with some more
details is to assess the sensitivity of water balance to rainfall seasonality. We refer to the Budyko framework (Budyko,
1974), which has been applied to relate water components in different climatic contexts worldwide, including the
Mediterranean climate (see e.g. Viola et al., 2017, Caracciolo et al. 2017). Specifically, the Budyko framework relates
the evaporative index (*ET$_a$/R*) to the dryness index (*ET$_p$/R*) computed at annual time scale in terms of "available water"





(i.e., rainfall $R$). Potential evapotranspiration, $ET_p$, is limited by either energy supply (for the dryness index less than or
equal to one) or water supply (for the dryness index greater than one) and therefore the Budyko space has two physical
bounds dictated by either the atmospheric water demand ($ET_a \leq ET_p$) or the atmospheric water supply ($ET_a \leq R$). The first
bound is the energy limit (or demand limit, i.e. the 1:1 line corresponding to $ET_a = ET_p$) implying that actual
evapotranspiration cannot exceed potential evapotranspiration. The second bound is the water limit (or supply limit, i.e.
the horizontal line corresponding to $ET_a = R$) implying that actual evapotranspiration cannot exceed precipitation when
dryness index is greater than one (i.e. $ET_p/R > 1$).
*Please insert Fig. 11 here*
By assuming that the long-term mean annual precipitation can be partitioned into the mean annual actual
evapotranspiration and mean annual water yield, according to the Budyko framework we assume that larger values of
dryness index (drier climate conditions) induce a greater proportion of rainfall that is partitioned to $ET_a$. In contrast,
data on the left-hand side of the Budyko curve will be characterized by a greater proportion of rainfall that is partitioned
to water yield. Fig. 11 shows the Budyko plot of dryness index ($ET_p/R$) versus evaporative index ($ET_a/R$) together with
the Budyko curve (solid garnet line). In this plot we have inserted the data points (colored dots) for the five different
durations of the rainy period in UARC obtained by the dynamic approach. A first comment is that all of these data
points gather within the energy-limited region of the Budyko plot, with the longest rainy period (blue dot) favoring
conditions of greater discharges (evaporative index of 0.45) and shortest rainy period (droughts indicated by the red dot)
inducing higher evapotranspiration fluxes (evaporative index of 0.54). This latter situation highlights that on average the
Upper Alento River catchment is characterized by a relatively good storage of soil-water made possible by the hydraulic
properties of the soils and the large portion of shrub spots and forest areas (mostly chestnut deciduous forests and olive
orchards), together with a good amount of annual precipitation in a hilly and mountainous zone of southern Italy.
However, $ET_p$ and $ET_a$ are not almost equivalent and one can even note that all of these data points cluster below the
Budyko curve (Williams et al., 2012). The observed departure below the Budyko curve can be due to a number of



reasons. Allowing for the Budyko assumptions for water balance, the present study refers to a long time scale (90
years), but a relatively small spatial scale since UARC has a drainage area of 102 km$^2$ and therefore local conditions
and controlling factors might exert some effects on the water budget calculations. Actually, rainfall seasonality (i.e.
intra-annual variability) can just be one of the major factors having led to a departure from the Budyko curve. The
typical Mediterranean climate, which is characterized by a precipitation being out-of-phase with potential
evapotranspiration, is also singled out as a cause of the deviations we have observed in our case study from the Budyko
curve (Milly, 1994). Normal situations, characterized by a wet season lasting 5-6 months (green dot), lead to partition
rainfall into 49% $ET_a$, as indicated by the evaporative index value of 0.49. We hereby recall that this study is based on
the assumption that the catchment response is not affected by human interferences and their feedbacks (land-use
change, change in soil hydraulic properties, enhanced evapotranspiration induced by global warming, etc.), but only by
changes in rainfall seasonality that, of course, can undermine Budyko's implicit assumption of temporal steady-state
(Feng et al., 2012; Troch et al., 2013).
*Please insert Fig. 12 here*
*Please insert Table 6 here*
The relationships between seasonal dryness index and water yield to rainfall ratio (*WY/R*) are affected by the duration of
the wet season and are depicted in Fig. 12. The coefficients of the exponential regression models with their
corresponding $R^2$-values pertaining to the wet or dry season are reported for each duration class of the rainy period in
Table 6. The exponential curves in the wet season (see plot 12a) are virtually parallel among them yielding, for a fixed
$ET_p/R$, more *WY/R* as the duration of the rainy period increases from 3-4 months to 7-8 months. In contrast, the
exponential regression curves belonging to the dry season (see plot 12b) are able to explain only a small amount of the
variations of *WY/R* in response to the dryness index and all seem quite insensitive to rainfall seasonality. Only the
exponential model pertaining to the dry season and for the smaller duration of the rainy period (3-4 months) explains a
bit less than 50% of the variability of $ET_p/R$ for the study catchment.




### 6. Conclusions

Capturing the relationship between rainfall and catchment-scale water balance components is a scientific challenge in
view of climate change in Mediterranean ecosystems. Water yield feeds a multi-use water reservoir in the ARC. This
study assesses rainfall seasonality by using two different approaches. The first one (static approach) is based on the
analysis of the SPI-values by identifying three seasonal features (a 4-month dry season, a 4-month rainy period, and two
2-month transition seasons). Seasonal anomalies are considered when the transition seasons turn into dry or wet season.
The second approach (dynamic approach) is based on the centroid and duration of the rainy period. In this study we
assumed the centroid as time-invariant while the temporal variability of the duration is described by a Gaussian
distribution. Rainfall seasonality was decomposed in seasonal duration, mean rainfall depth and rainfall frequency. The
impact of seasonality anomalies on water balance components was evaluated in both approaches by providing simulated
water yield, actual evapotranspiration and groundwater recharge within a probabilistic framework. The seasonal
anomalies occur on the tails of the normal distribution. Both approaches concur on the impact of rainfall seasonal
anomalies on catchment-scale water balance components. A drought anomaly (prolonged duration of the dry season)
potentially leads to a decrease of about 20% in annual average rainfall inducing a decline of about 27%, 10% and 34%
of annual average amounts of water yield, actual evapotranspiration and groundwater recharge, respectively. An
exceptional prolonged wet season will cause an increase of about 13% in annual average rainfall inducing a rise of
about 21%, 3% and 28% of annual average amounts of water yield, actual evapotranspiration and groundwater
recharge, respectively.
In the dynamic approach, we demonstrated that the implicit assumption of temporal steady-state in the Budyko relation
approach is quite sensitive to rainfall seasonality. The Budyko evaporative index spans from 0.45 to 0.54 when wet
season lasts from 7-8 months up to 3-4 months. Moreover, it is possible to identify distinct seasonal-dependent
regression equations linking seasonal water yield to dryness index over the wet season.



A subsequent study will integrate the discussion on water supply with projected water consumption in the next decades
induced by socio-economic controls and climate variability. The challenge is to forecast extreme drought episodes in
consecutive years that might lead to plausible water crisis at the water reservoir.

**7. Appendix**
We set $k$ and $m$ as counters for the hydrological year and the 12 months in each year, respectively.
The annual rainfall, $R_k$ and associated monthly probability distribution, $p_{k,m}$ are defined as:
$R_k = \sum_{m=1}^{12} r_{k,m}$ (A1)
$p_{k,m} = \dfrac{r_{k,m}}{R_k}$ (A2)
where $r_{k,m}$ represents the rainfall depth recorded in the $m$-th month in the $k$-th year.
The relative entropy, $D_k$ is calculated in each hydrological year $k$, as:
$D_k = \sum_{m=1}^{12} p_{k,m} \, log_2 \left( \dfrac{p_{k,m}}{q_m} \right)$ (A3)
where $q_m$ is equal to 1/12 (uniform distribution). This statistical index quantifies the distribution of monthly rainfall
within each hydrological year. Finally, the dimensionless seasonality index ($DSI_k$) in each hydrological year $k$, is given
by:
$DSI_k = D_k \dfrac{R_k}{\bar{R}_{max}}$ (A4)
where $\bar{R}_{max}$ is maximum $\bar{R}$. This way $DSI_k$ is zero when rainfall is uniformly distributed throughout the year and
reaches its maximum value $log_2 12$ when rainfall is concentrated in a single month.



According to Feng et al. (2013), the magnitude ($R_k$) represents annual rainfall whereas the centroid ($C_k$) and the spread
($Z_k$) indicate timing and duration of the wet season, respectively and are calculated in each hydrological year $k$ as:
$C_k = \frac{1}{R_k}\sum_{m=1}^{12} m r_{k,m}$                                                                            (A5)
$Z_k = \sqrt{\frac{1}{R_k}\sum_{m=1}^{12}|m - C_k|^2 r_{k,m}}$                                                             (A6)

**Acknowledgments**
The study reported in this paper was partially supported by the MiUR-PRIN Project "Innovative methods for water
resources management under hydro-climatic uncertainty scenarios" (grant 2010JHF437). The Director of the
"Consorzio di Bonifica Velia", Marcello Nicodemo, is also acknowledged for his support in providing the datasets
recorded at the Piano della Rocca earth dam. Roberto Deidda acknowledges the financial support received from the
Sardinia Region under grant L.R. 7/2007, funding call 2017, CUP: F76C18000920002

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





597        **Figures**



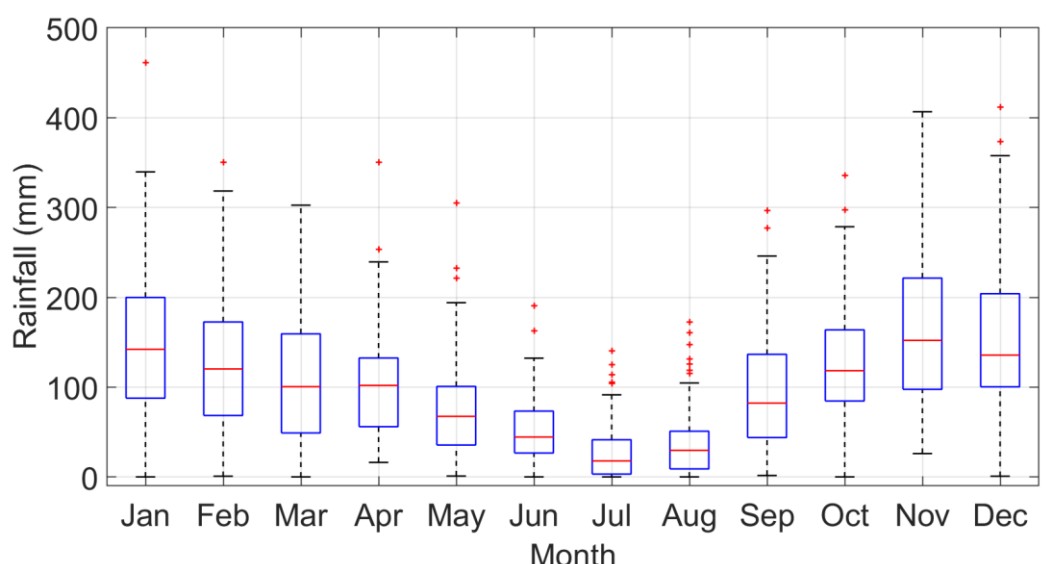


**Figure 1: Box plots of the monthly rainfall depths recorded at the Gioi Cilento weather station (1920-2018).**

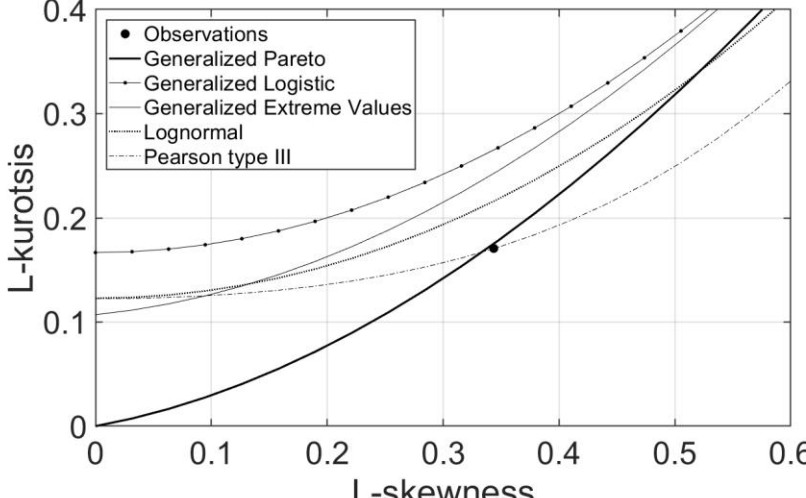


**Figure 2: Theoretical L-moment ratio of common distribution models, as compared to the sample L-moment ratios of daily**
**rainfall time series at the Gioi Cilento weather station (filled large circle).**








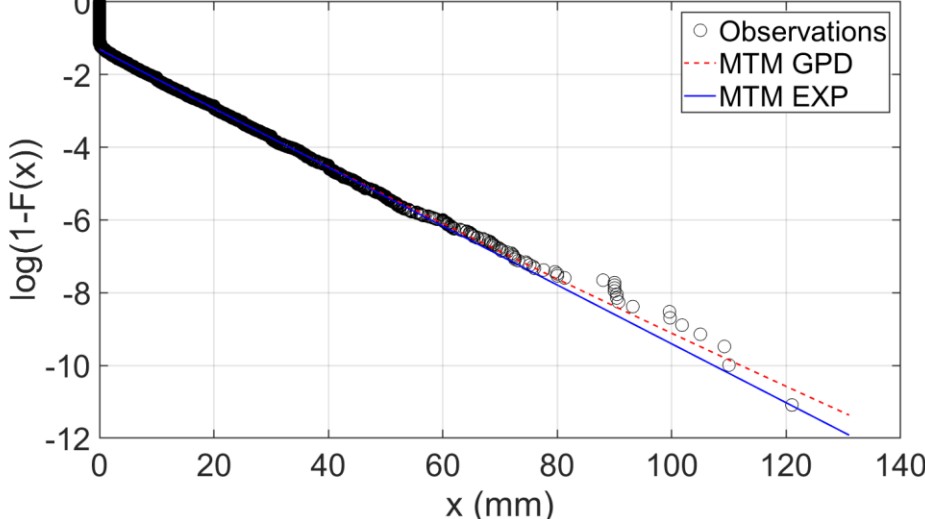


**Figure 3: Exponential probability plot of empirical and fitted cumulative distribution functions of daily rainfall depths collected at the Gioi Cilento weather station.**





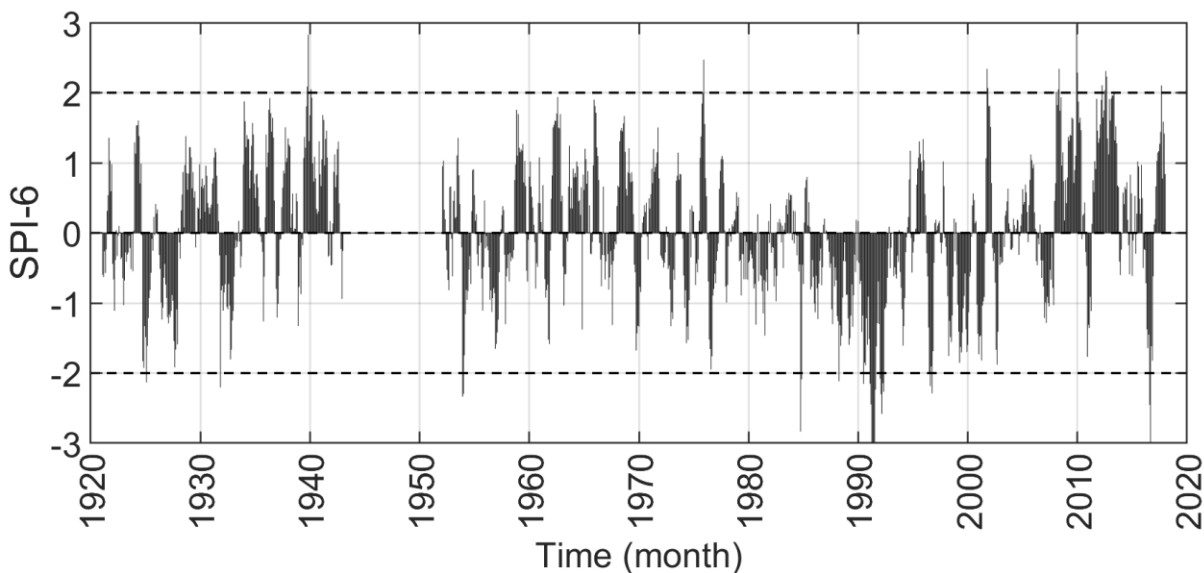

**Figure 4: Temporal evolution of SPI-6 spanning from 1920 to 2018 (rainfall data were recorded at the Gioi Cilento weather station).**

620



Figure 5: a) Temporal evolution of SPI-values (gray bars) and their 12-month moving average (magenta line) spanning from 1920 to 2018 in the static approach; b) Box plots of SPI-values and frequency distribution in the c) rainy period (blue histograms corresponding to Nov-Dec-Jan-Feb), d) transition period (yellow histograms corresponding to Mar-Apr-Sep-Oct), e) dry period (red histograms corresponding to May-Jun-Jul-Aug).









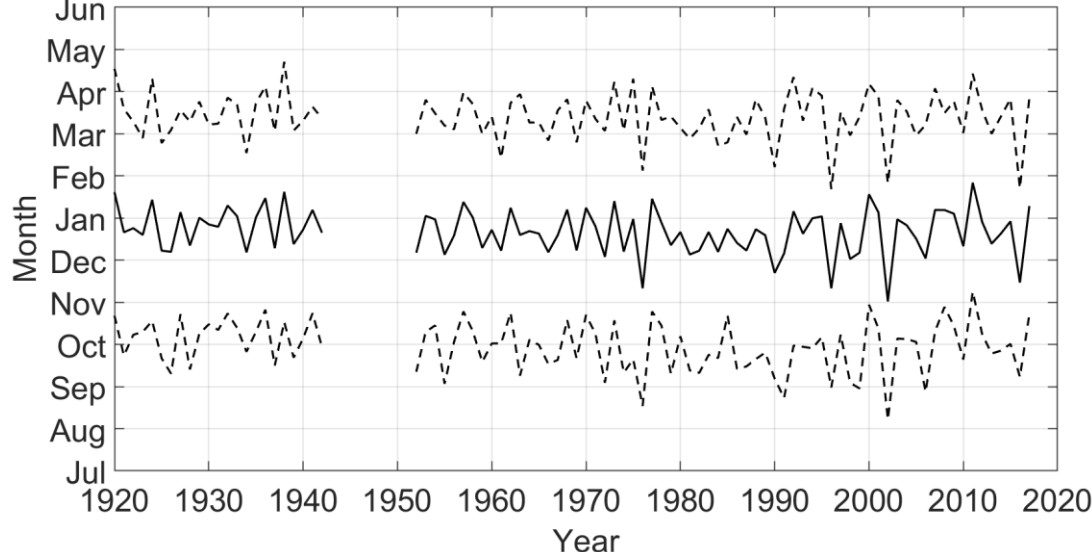


**Figure 6: Temporal trend of the centroid, or timing (solid line), and spread, or duration (dashed lines) of the monthly rainfall distribution spanning from 1920 to 2018 in the dynamic approach (rainfall data were recorded at the Gioi Cilento weather station).**

635   .








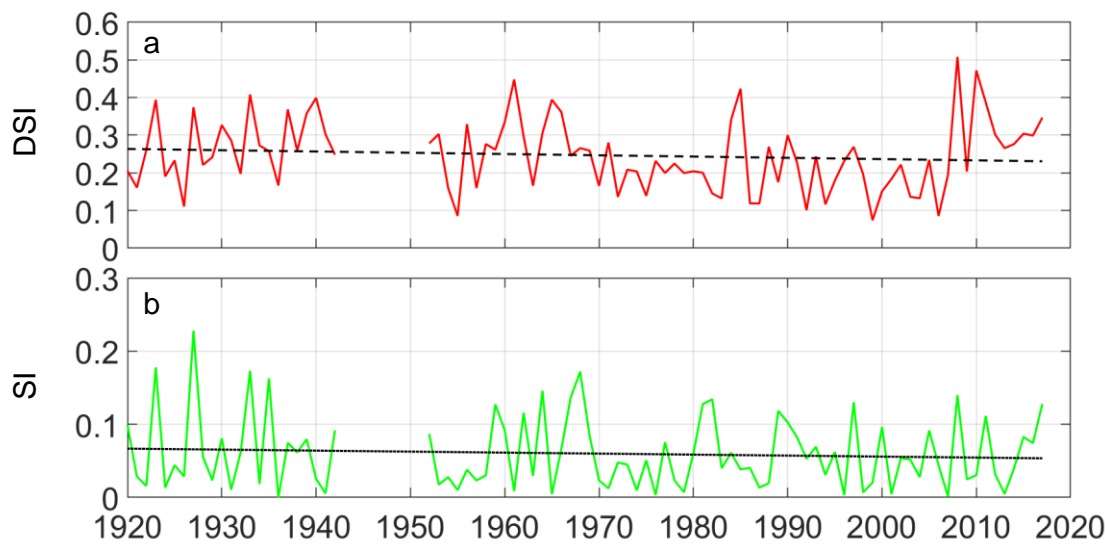


**Figure 7: Temporal evolution of a) dimensionless seasonal index, DSI (Feng et al., 2013) represented by a red line with corresponding linear regression (dashed line); b) seasonality index, SI (Walsh and Lawler, 1981) represented by a green line with corresponding linear regression (dotted line).**











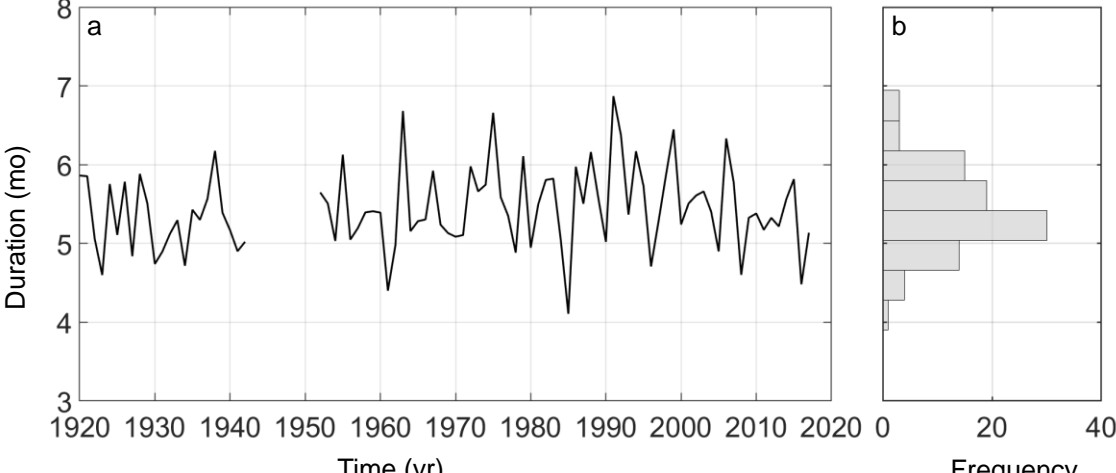


**Figure 8: Time series (a) and frequency distribution (b) of durations of the rainy periods at the Gioi Cilento weather station in the dynamic approach.**





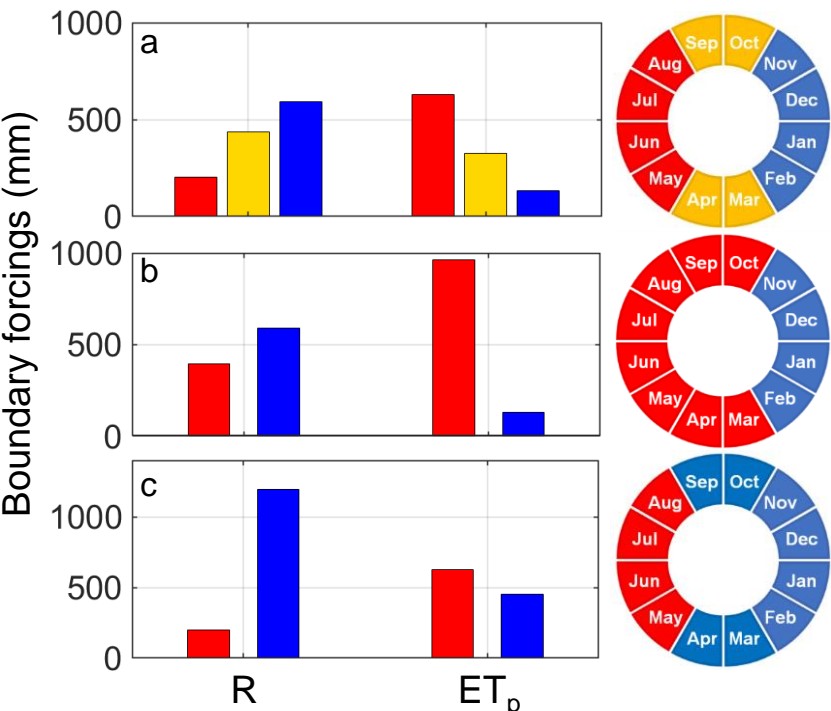

**Figure 9: Boundary forcings in the static approach, namely seasonal rainfall ($R$) and potential evapotranspiration ($ET_p$) in the dry (red bars), transition (orange bars), and wet season (blue bars). Three scenarios are presented: a) "reference scenario" with the dry, transition, and wet seasons all lasting 4 months; b) "dry scenario" with the dry and wet seasons lasting 8 and 4 months, respectively; c) "wet scenario" with the dry and wet seasons lasting 4 and 8 months, respectively.**

662

663

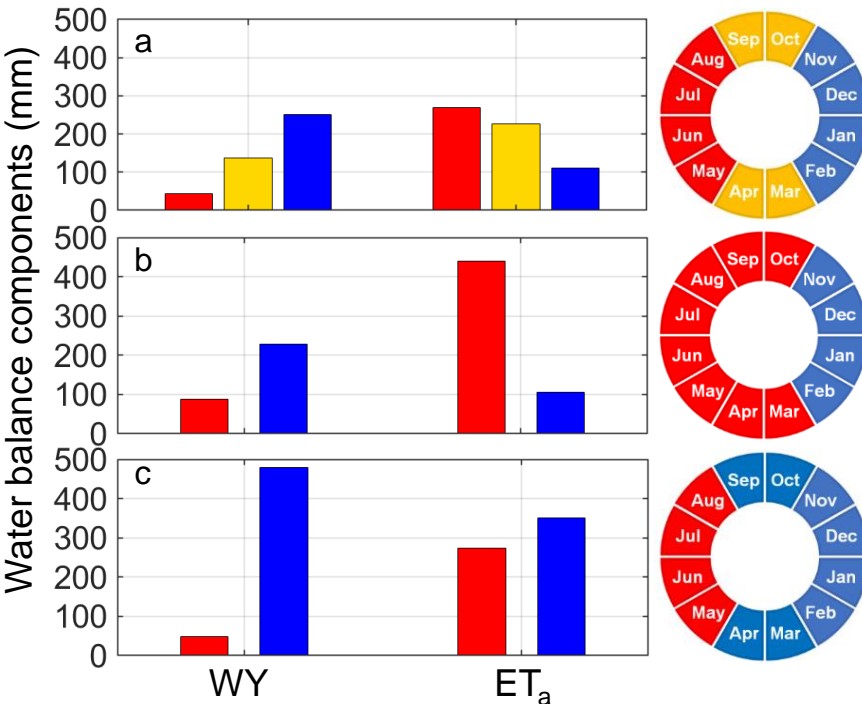

**Figure 10: Main water balance components in the static approach, namely seasonal water yield (WY) and actual evapotranspiration ($ET_a$) in the dry (red bars), transition (orange bars), and wet season (blue bars). Three scenarios are presented: a) "reference scenario" with the dry, transition, and wet seasons all lasting 4 months; b) "dry scenario" with the dry and wet seasons lasting 8 and 4 months, respectively; c) "wet scenario" with the dry and wet seasons lasting 4 and 8 months, respectively.**

670



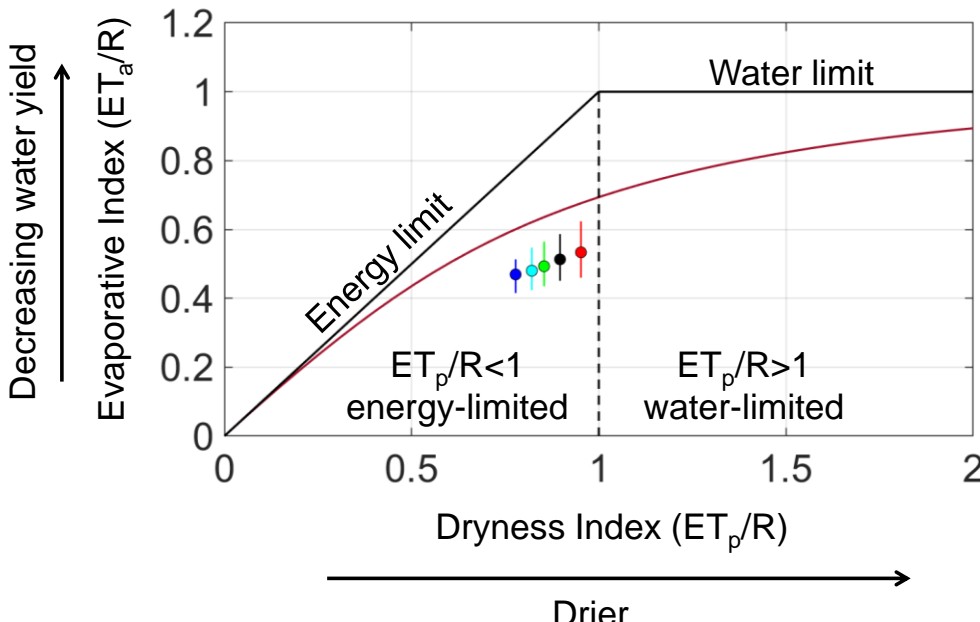

671

**Figure 11: Budyko diagram relating dryness index ($ET_p/R$) with evaporative ($ET_a/R$) index classified according to the duration of the rainy period pertaining to the dynamic approach. Circles denote median and vertical colored lines represent the range between 5[th] and 95[th] percentiles of evaporative index (red, black, green, cyan and blue colors correspond to duration of the rainy period of 3-4, 4-5, 5-6, 6-7 and 7-8 months, respectively). Solid lines denote energy and water limits, solid garnet line represents the Budyko curve (Budyko, 1974). The vertical dashed line separates left-hand side from right-hand side of the Budyko curve.**











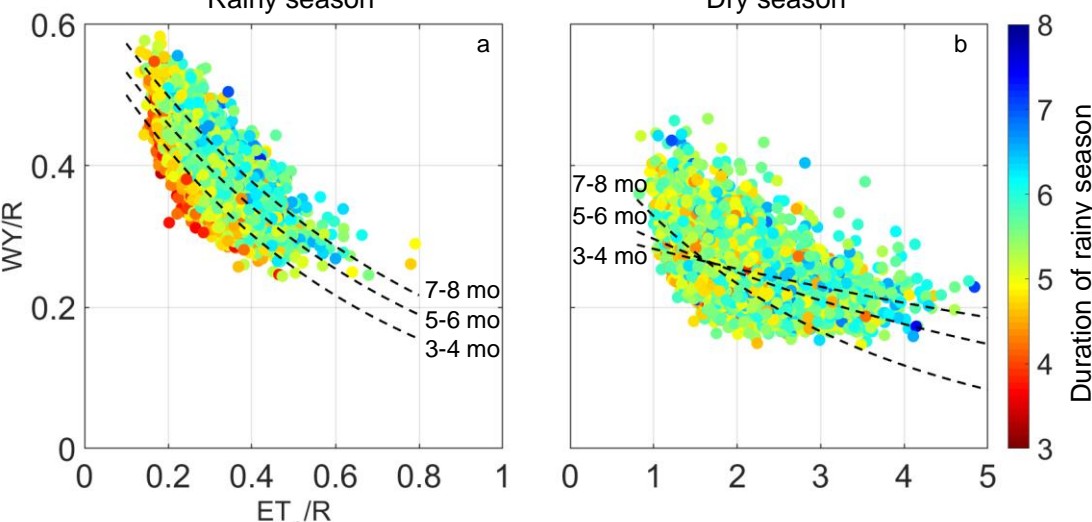


**Figure 12: Relationship between dryness index and water yield to rainfall ratio (*WY/R*) on seasonal basis and classified according to the duration of the wet season (from shortest to longest denoted by reddish and bluish colors in the colorbar) pertaining to the dynamic approach for the wet season (plot 12a) and the dry season (plot 12b). The exponential regression equations are represented in both plots by the dashed black lines according to the duration of the rainy period.**







**Tables**


**Table 1: Descriptive statistics of the monthly rainfall distributions recorded at the Gioi Cilento weather station**
**during the period 1920-2018.**

| month | mean | median | min | max | Std. Dev. | CV |
|---|---|---|---|---|---|---|
| | mm | mm | mm | mm | mm | % |
| Jan | 145.6 | 141.65 | 0.0 | 461.2 | 81.6 | 56.0 |
| Feb | 128.1 | 120.25 | 0.8 | 350.1 | 76.3 | 59.6 |
| Mar | 112.9 | 101.1 | 0.0 | 302.6 | 73.4 | 65.0 |
| Apr | 102.5 | 101 | 16.2 | 350.6 | 59.5 | 58.0 |
| May | 75.2 | 67.6 | 1.1 | 304.8 | 56.6 | 75.2 |
| Jun | 52.8 | 45.3 | 0.0 | 190.9 | 38.2 | 72.3 |
| Jul | 29.8 | 17.6 | 0.0 | 140.4 | 32.8 | 110.0 |
| Aug | 39.7 | 30.3 | 0.0 | 210 | 42.8 | 107.7 |
| Sep | 94.4 | 81.9 | 1.6 | 296.8 | 63.0 | 66.7 |
| Oct | 126.8 | 118.8 | 0.0 | 335.5 | 70.3 | 55.4 |
| Nov | 166.9 | 152.2 | 26.0 | 613.2 | 94.9 | 56.9 |
| Dec | 154.6 | 134.55 | 0.8 | 411.8 | 85.1 | 55.1 |


**Table 2: Scenario set up in the "*static*" approach. Duration and Poisson distribution parameters ($\eta$ and $\lambda$) are**
**reported for each of the considered scenarios.**

| | Dry season | | | Transition season | | | Wet season | | |
|---|---|---|---|---|---|---|---|---|---|
| | months | $\eta$ | $\lambda$ | months | $\eta$ | $\lambda$ | months | $\eta$ | $\lambda$ |
| | - | mm | d$^{-1}$ | - | mm | d$^{-1}$ | - | mm | d$^{-1}$ |
| Reference scenario (static) | 4 | 8.20 | 0.196 | 4 | 10.53 | 0.34 | 4 | 11.70 | 0.423 |
| Dry scenario (static) | 8 | 8.20 | 0.196 | 0 | - | - | 4 | 11.70 | 0.423 |
| Wet scenario (static) | 4 | 8.20 | 0.196 | 0 | - | - | 8 | 11.70 | 0.423 |







**Table 3: Scenario set up in the "*dynamic*" approach. Duration and Poisson distribution parameters ($\eta$ and $\lambda$) are**
**reported in the dry and wet season.**

| Dynamic scenario | Dry season | | | Wet season | | |
|---|---|---|---|---|---|---|
| | months | $\eta$ | $\lambda$ | months | $\eta$ | $\lambda$ |
| | - | mm | $d^{-1}$ | - | mm | $d^{-1}$ |
| | random | 9.34 | 0.243 | random | 11.99 | 0.413 |



**Table 4: Descriptive statistics of annual water balance components obtained in the three scenarios**
**in the "*static*" approach. Units are mm, except for CV (%).**

| Scenario | Variable | $R$ | $WY$ | $ET_a$ | $GR$ |
|---|---|---|---|---|---|
| | | mm | mm | mm | mm |
| Reference scenario | mean | 1229.0 | 433.3 | 605.2 | 194.3 |
| | stand. dev. | 176.0 | 104.2 | 36.5 | 48.0 |
| | CV (%) | 14.3 | 24.1 | 6.0 | 24.7 |
| | min | 586.6 | 150.8 | 449.1 | 44.0 |
| | max | 2053.9 | 1005.9 | 743.0 | 389.6 |
| Dry scenario | mean | 987.7 | 317.3 | 545.1 | 128.0 |
| | stand. dev. | 155.5 | 88.1 | 40.8 | 42.7 |
| | CV (%) | 15.7 | 27.8 | 7.5 | 33.4 |
| | min | 498.7 | 96.2 | 396.0 | 7.2 |
| | max | 1649.9 | 802.4 | 691.6 | 319.3 |
| Wet scenario | mean | 1392.8 | 526.0 | 625.8 | 248.1 |
| | stand. dev. | 192.4 | 119.6 | 34.3 | 52.6 |
| | CV (%) | 13.8 | 22.7 | 5.5 | 21.2 |
| | min | 721.9 | 157.0 | 481.2 | 59.0 |
| | max | 2179.2 | 1088.2 | 748.6 | 461.6 |








**Table 5: Water balance components associated to occurrence probabilities for each duration of the rainy period.**

|  | Probability | $R$ | $WY$ | $ET_a$ | $GR$ |
|---|---|---|---|---|---|
|  | % | mm | mm | mm | mm |
| 3-4 months | 0.6% | 1,145.0 | 385.3 | 608.5 | 169.6 |
| 4-5 months | 23% | 1,213.4 | 420.0 | 619.4 | 188.0 |
| 5-6 months | 62% | 1,275.4 | 453.0 | 624.9 | 199.6 |
| 6-7 months | 14% | 1,326.0 | 480.2 | 631.6 | 210.2 |
| 7-8 months | 0.3% | 1,383.5 | 511.6 | 644.2 | 211.8 |


**Table 6: Exponential regression models, with the corresponding coefficient of determination ($R^2$), for the wet and dry seasons as a function of the duration of the rainy period.**

| Duration | Wet season | | Dry season | |
|---|---|---|---|---|
|  | Exp regression function | $R^2$ | Exp regression function | $R^2$ |
| 3-4 months | $WY/R = 0.5914 \times \exp(-1.674 \times ET_p/R)$ | 0.440 | $WY/R = 0.4635 \times \exp(-0.343\ ET_p/R)$ | 0.482 |
| 4-5 months | $WY/R = 0.6031 \times \exp(-1.536 \times ET_p/R)$ | 0.579 | $WY/R = 0.3675 \times \exp(-0.204 \times ET_p/R)$ | 0.290 |
| 5-6 months | $WY/R = 0.6171 \times \exp(-1.477 \times ET_p/R)$ | 0.587 | $WY/R = 0.3530 \times \exp(-0.174 \times ET_p/R)$ | 0.279 |
| 6-7 months | $WY/R = 0.6313 \times \exp(-1.399 \times ET_p/R)$ | 0.617 | $WY/R = 0.3476 \times \exp(-0.159 \times ET_p/R)$ | 0.284 |
| 7-8 months | $WY/R = 0.6586 \times \exp(-1.389 \times ET_p/R)$ | 0.585 | $WY/R = 0.3137 \times \exp(-0.105 \times ET_p/R)$ | 0.211 |

