# Peer review of "Assessing the impact of rainfall seasonality anomalies on catchment-scale water balance components"

_Hydrology and Earth System Sciences, 2019_

## Referee Comment (RC1) · Anonymous Referee #1 · 15 Nov 2019

General comments:

In the submitted paper, authors investigate the impact of the rainfall seasonality anomalies on the catchment water balance components. For this purpose, a catchment in the southern Italy is selected and SWAT model is applied in order to carry out the investigation. Two different approaches are used in order to define rainfall scenarios. First approaches is based on the standardized precipitation index and second one is based on the duration of the wet season as proposed by Feng et al. (2013). The topic is potentially interesting for the society and HESS readers. However, two main shortcomings of the paper from my perspective that should be improved are:

Firstly, the main focus of the paper is to investigate what is the impact of different rainfall scenarios on the water yield, actual evapotranspiration and groundwater recharge.

Thus, for different scenarios changes in these variables are analyzed with respect to reference case. Model calibration is just briefly described and reference to more detailed description is given (Nesta et al., 2017). It seems that model was calibrated using monthly data (?). However, P6, L124 states that daily time step of the SWAT model was used. I think that model should also be calibrated using daily data if authors want to use this time step. Otherwise, I would suggest to aggregate daily rainfall data into monthly and re-run the model with monthly time step (if this is possible or perhaps use a different model). An alternative is, to calibrate the model using daily data if there is a discharge gauging station available near the catchment outlet.

Secondly, when using different scenarios, authors only modified rainfall characteristics, what about air temperature? It is true that in some cases the dependence between these two variables can be low or even none existing. However, is some other cases, some dependence could exist. For example, higher average annual temperature could lead to lower annual rainfall and vice-versa. Or higher daily temperature in summer could cause higher rainfall amounts due to more extreme thunderstorm. Did authors check the relationship for this specific catchment? Moreover, I think that air temperature variability should be included in this kind of investigations. Even if there is no clear relationship with rainfall.

Specific comments:

I would suggest to add a figure showing the location of the catchment with stations used.

P6, L130: Please better explain what is meant by the term boundary forcings.

P7, L142-144: Why did you used only 3 years for simulation and why 2-years warm-up period? How does this selection impact on the results? Moreover, does initial state of the catchment also has impact on the results (i.e. using different initial values of model variables)?

P7, L146-149: The data from other station will be used for analyses at monthly time scale but the model will run with daily time step and daily reference evapotranspiration will be calculated? Perhaps you could rephrase this sentence.

P7, L149: Here reference evapotranspiration is mentioned but in next sections, you only mention potential and actual evapotranspiration. Why was reference evapotranspiration used?

P9, section 4.3: If I understand correctly exponential distribution was selected only based on the graphical comparison shown in Figure 2 and Figure 3? If this is the case, I would suggest to additionally apply a suitable statistical test.

P10, L230-231: I do not understand this sentence, if you split the data, how can you then have a drying trend? Only for the second 45 years?

P13, L285 and L294: A statistical test is mentioned here but no information about null and alternative hypothesis is given. Moreover, authors should rephrase these sentences. In statistical hypothesis testing the null hypothesis can be either rejected in favor of the alternative hypothesis or cannot be rejected (with the chosen significance level). Moreover, all the methods used should probably be mentioned and described in the methodology section (and not results and discussion).

Sections 5.3 and 5.4 and conclusions: The main results of the paper are somehow expected: dry scenario leads to less runoff, groundwater recharge and also less actual evapotranspiration (compared to reference scenario). On other hand, wet scenario leads to more runoff, groundwater recharge and actual evapotranspiration (compared to reference scenario). Moreover, different rainfall simulation methods yield different results. The actual relationship among variables mostly depends on the rainfall characteristics, especially if variability in air temperature is not considered. Can the authors perhaps somehow enhance the take home message of this paper?

[Figure]

553, 2019.

---

## Referee Comment (RC2) · Anonymous Referee #2 · 25 Nov 2019

The paper deals with the assessment of water balance components (i.e. water yield, evapotranspiration, groundwater recharge, etc) and relative deficit in case of climatic anomalies related to seasonality in a Mediterranean basin. This is done by parameterizing a rainfall generator model according to two different schematic representation of seasonality (called "static" and "dynamic"), and using synthetic rainfall series as input to the SWAT hydrological model.

While shifts and changes in seasonal patterns have been addressed by many researchers as key factors in analyzing the hydrological impact of climatic fluctuations, the consequent issue of how these phenomena may impact the regulation of artificial reservoirs, designed for annual or multiyear storage purpose, deserves attention.

The paper is in general well sounded and relevant although it could be improved in my

opinion, accounting for the following suggestions:

1. The paper is compound by two main issues: the first one is referred to the analysis of the climatic forcing and the parameterization of the rainfall model; the second one is related to the use of SWAT model to obtain different components of water balance. A stronger emphasis is given to the first one, which is also performed by comparing different methods, while the second one is much less discussed. Also, the overall paper goal could be better assessed and the methodology more detailed in the introduction. To make an example, the sentence "The goal of the study is to characterize the rainfall seasonality and its anomalies by using two approaches." (line 81) is in my opinion somehow misleading with respect to the overall paper objectives and developments.

2. Dealing with issue #1, i.e. seasonality assessment, in the introduction the PCI and SI methods are indicated as most popular approaches. Nevertheless, the authors do not use them but rather prefer an SPI based analysis and the procedure proposed by Feng et al (2013). A better acknowledgement could be provided about the reasons of such choices, and the comparisons between the performances of different methods.

3. AT line 184 the authors state that they "assumed that the duration of the wet season follows a normal distribution...". While I do not doubt that such hypothesis may be a feasible one, I would expect some kind of validation or testing of it through observed data.

4. The stochastic Poisson point process with exponential distribution of pulses that is finally used for rainfall generations, I believe could be referenced to classical papers like Rodriguez-Iturbe, I. et al (Journal of Geophysical Research, 1987) and /or Eagleson (WRR, 1972), may be also of interest a more recent application by Veneziano and Iacobellis (WRR, 2002) on Italian datasets, among many others. The use of seasonal parameterization on a stochastic rainfall generator is also a matter of interest.

5. I believe that also conclusions should be reinforced. First by better depicting which practical use the methodology could be exploited for and, second, by deepening the
discussion about the characterization of rainfall seasonality and its anomalies, according to different approaches, which was mentioned as a goal of the study.

---

## Author Comment (AC1) · 27 Nov 2019

**Reply Referee #1**

COMMENT 1: General comments: In the submitted paper, authors investigate the impact of the rainfall seasonality anomalies on the catchment water balance components. For this purpose, a catchment in the southern Italy is selected and SWAT model is applied in order to carry out the investigation. Two different approaches are used in order to define rainfall scenarios. First approaches is based on the standardized precipitation index and second one is based on the duration of the wet season as proposed by Feng et al. (2013). The topic is potentially interesting for the society and HESS readers.

REPLY-1: We thank this reviewer for her/his comments. In the following sections we have tried to provide some preliminary replies to clarify her/his major concerns.

However, two main shortcomings of the paper from my perspective that should be improved are:

COMMENT 2: Firstly, the main focus of the paper is to investigate what is the impact of different rainfall scenarios on the water yield, actual evapotranspiration and groundwater recharge. Thus, for different scenarios changes in these variables are analyzed with respect to reference case. Model calibration is just briefly described and reference to more detailed description is given (Nesta et al., 2017). It seems that model was calibrated using monthly data (?). However, P6, L124 states that daily time step of the SWAT model was used. I think that model should also be calibrated using daily data if authors want to use this time step. Otherwise, I would suggest to aggregate daily rainfall data into monthly and re-run the model with monthly time step (if this is possible or perhaps use a different model). An alternative is, to calibrate the model using daily data if there is a discharge gauging station available near the catchment outlet.

REPLY-2: Nasta et al. (2017 STotEnv) calibrated a few model parameters by comparing measured and simulated monthly water yield values recorded at the dam. Numerical simulations were run at daily time step (the only time step allowed in SWAT). In this study, we followed the same criterion, we run numerical simulations at daily time step (rainfall was randomly generated at daily time step) and aggregated the output fluxes at monthly time resolution. We are aware that calibrating at monthly time-scale might lead to a potential misfit between measured and simulated values at daily time-scale (e.g. Adla et al., 2019, Water). However, our analysis is based on the monthly aggregation of fluxes and we analyzed seasonal patterns of monthly aggregates. This important point will be clarified in the revised manuscript.

Adla, S., S. Tripathi, M. Disse, 2019. Can we calibrate a daily time-step hydrological model using monthly time-step discharge data? Water 11, 1750; doi:10.3390/w11091750.

COMMENT 3: Secondly, when using different scenarios, authors only modified rainfall characteristics, what about air temperature? It is true that in some cases the dependence between these two variables can be low or even none existing. However, is some other cases, some dependence could exist. For example, higher average annual temperature could lead to lower annual rainfall and vice-versa. Or higher daily temperature in summer could cause higher rainfall amounts due to more extreme thunderstorm. Did authors check the relationship for this specific catchment? Moreover, I think that air temperature variability should be included in this kind of investigations. Even if there is no clear relationship with rainfall.

REPLY-3: We fully agree with this comment, which in our opinion is timely. Indeed, the feedback of temperature on precipitation is a widely recognized cause for the increasing frequency of storms and heavy rainfall events under climate change. Unfortunately, we do not have easy access to database of long time series of temperature measurements to reliably evaluate this linkage in our case study, since this analysis cannot be restricted only to the surrounding air temperatures, but should include the change in sea surface temperature and air moisture to correctly frame the analysis. This is certainly an issue that deserves an in-depth analysis, but at this stage it goes beyond the scope of our analysis. We have focused our study only on the long-term (almost one century) daily timeseries of rainfall, yielding interesting outcomes about the sensitivity of catchment hydrological response to seasonal rainfall patterns only. Moreover, for the same reason (lack of temperature data) we could not capture any significant (increasing) trend and assumed temperature stationary in time and so does evapotranspiration. Our analysis addressed only on the impact exerted by rainfall seasonality on water balance components. The availability of adequate datasets should

also highlight whether the observed increases in air temperatures impact the daily precipitations, or mostly only the sub-daily and sub-hourly rainfall values.

Specific comments:
I would suggest to add a figure showing the location of the catchment with stations used.
REPLY-4: Actually, the figure with the locations of the two weather stations is already showed in Nasta et al. (2017), so we can easily refer to it.

P6, L130: Please better explain what is meant by the term boundary forcings.
REPLY-5: With the term "boundary forcings", we mean the (input) water fluxes (rainfall and potential evapotranspiration) set on the upper boundary condition of the soil domain.

P7, L142-144: Why did you used only 3 years for simulation and why 2-years warm-up period? How does this selection impact on the results? Moreover, does initial state of the catchment also has impact on the results (i.e. using different initial values of model variables)?
REPLY-6: We decided to run 3 years in each scenario and neglect the model simulations of the first two years to annihilate the impact of initial soil moisture values set in the soil domain. We point out that the soil water content at the initial day of year 1 is set at field capacity (which can be already considered a realistic situation in winter under Mediterranean climate). We also specify that we consider the third year of model simulation. We repeat this exercise 10,000 times such that to organize output fluxes in a probabilistic framework.

P7, L146-149: The data from other station will be used for analyses at monthly time scale but the model will run with daily time step and daily reference evapotranspiration will be calculated? Perhaps you could rephrase this sentence.
REPLY-7: We agree and we reformulate this (unclear) sentence. We have the daily weather values and we use the descriptive statistics of daily values in each month of the year to generate new random daily values of evapotranspiration in each month.

P7, L149: Here reference evapotranspiration is mentioned but in next sections, you only mention potential and actual evapotranspiration. Why was reference evapotranspiration used?
REPLY-8: SWAT uses the weather data to estimate potential evapotranspiration ($ET_p$). We will replace the word "reference" with the word "potential" to avoid confusion. Thanks for pointing it out.

P9, section 4.3: If I understand correctly exponential distribution was selected only based on the graphical comparison shown in Figure 2 and Figure 3? If this is the case, I would suggest to additionally apply a suitable statistical test.
REPLY-9: The stochastic Poisson point process of daily rainfall occurrences was assumed to represent daily rainfall evolution for its easy reproducibility (L193-196). In a preliminary analysis we tested it and compared it with the best parent distribution, namely the Generalized Pareto Distribution. In this case, we observed a fair agreement between the two models for representing daily rainfall evolution recorded at Gioi Cilento weather station and concluded that the simple-to-use exponential model is suitable (L206-210).
The stochastic Poisson point process is widely used for its simplicity and parsimony as pointed out by Reviewer#2 (we list below the three references reported in REPLY-4 to ref#2)

Rodríguez-Iturbe, I., B. Febres de Power, J.B. Valdés. 1987. Rectangular pulses point process models for rainfall: Analysis of empirical data. Journal of Geophysical Research, https://doi.org/10.1029/JD092iD08p09645

Veneziano, D., V. Iacobellis. 2002. Multiscaling pulse representation of temporal rainfall. Water Resources Research, 38, 1138, 10.1029/2001WR000522

Eagleson, P. S. 1972. Dynamics of flood frequency. Water Resour. Res.,8, 878–898.

P10, L230-231: I do not understand this sentence, if you split the data, how can you then have a drying trend? Only for the second 45 years?

REPLY-10: The frequency distribution of SPI-6 values in the first 45 years is wetter than the one pertaining to the last (more recent) 45 years.

P13, L285 and L294: A statistical test is mentioned here but no information about null and alternative hypothesis is given. Moreover, authors should rephrase these sentences. In statistical hypothesis testing the null hypothesis can be either rejected in favor of the alternative hypothesis or cannot be rejected (with the chosen significance level). Moreover, all the methods used should probably be mentioned and described in the methodology section (and not results and discussion).

REPLY-11: Mann-Kendal test has become a standard test for trend under climate change, it is widely applied in many papers. Therefore, after a discussion among the co-authors, we decided not to describe the Mann-Kendal test in detail and just mentioned the reference.

Sections 5.3 and 5.4 and conclusions: The main results of the paper are somehow expected: dry scenario leads to less runoff, groundwater recharge and also less actual evapotranspiration (compared to reference scenario). On other hand, wet scenario leads to more runoff, groundwater recharge and actual evapotranspiration (compared to reference scenario). Moreover, different rainfall simulation methods yield different results. The actual relationship among variables mostly depends on the rainfall characteristics, especially if variability in air temperature is not considered. Can the authors perhaps somehow enhance the take home message of this paper?

REPLY-12: The target of our study is to evaluate the sensitivity of water balance components to rainfall seasonality only (potential temperature effects are not considered here, partly because of lack of suitable datasets). The take-home message of this paper is that allowing for rainfall seasonality is an important issue because the impact of seasonal duration can be seen in the Budyko plot (Fig. 11). Moreover the assumption of steady-state condition inherent the Budycko approach is questioned. The stationarity/non-stationarity dilemma in hydrological processes is still matter of an open debate in the scientific community (Milly et al., 2008; Montanari and Koutsoyiannis, 2014).

Milly, P. C. D., J. Betancourt, M. Falkenmark, R. M. Hirsch, Z. W. Kundzewicz, D. P. Lettenmaier, and R. J. Stouffer. 2008. Stationarity is dead: Whither water management?,Science,319, 573–574, doi:10.1126/science.1151915

Montanari, A., and D. Koutsoyiannis. 2014. Modeling and mitigating natural hazards: Stationarity is immortal!, Water Resour. Res.,50,9748–9756, doi:10.1002/2014WR016092.

---

## Author Comment (AC2) · 28 Nov 2019

**Reply Referee #2**

The paper deals with the assessment of water balance components (i.e. water yield, evapotranspiration, groundwater recharge, etc.) and relative deficit in case of climatic anomalies related to seasonality in a Mediterranean basin. This is done by parameterizing a rainfall generator model according to two different schematic representation of seasonality (called "static" and "dynamic"), and using synthetic rainfall series as input to the SWAT hydrological model. While shifts and changes in seasonal patterns have been addressed by many researchers as key factors in analyzing the hydrological impact of climatic fluctuations, the consequent issue of how these phenomena may impact the regulation of artificial reservoirs, designed for annual or multiyear storage purpose, deserves attention.

GENERAL REPLY: We thank this reviewer for the comments and suggestions. In the following sections we have tried to provide a few preliminary replies to clarify her/his major concerns.

The paper is in general well sounded and relevant although it could be improved in my opinion, accounting for the following suggestions. The paper is compound by two main issues:

1. the first one is referred to the analysis of the climatic forcing and the parameterization of the rainfall model; the second one is related to the use of SWAT model to obtain different components of water balance. A stronger emphasis is given to the first one, which is also performed by comparing different methods, while the second one is much less discussed. Also, the overall paper goal could be better assessed and the methodology more detailed in the introduction. To make an example, the sentence "The goal of the study is to characterize the rainfall seasonality and its anomalies by using two approaches." (line 81) is in my opinion somehow misleading with respect to the overall paper objectives and developments.
   REPLY-1: We agree with this concern that was partly raised by Reviewer#1. In the introduction we will try to give more emphasis to the sensitivity analysis of watershed hydrological response to rainfall seasonality. As stated in the reply to Reviewer#1, the take-home message is that one should account for rainfall seasonality because it might potentially undermine the hypothesis of steady-state condition in Budyko's approach.

2. Dealing with issue #1, i.e. seasonality assessment, in the introduction the PCI and SI methods are indicated as most popular approaches. Nevertheless, the authors do not use them but rather prefer an SPI based analysis and the procedure proposed by Feng et al (2013). A better acknowledgement could be provided about the reasons of such choices, and the comparisons between the performances of different methods.
   REPLY-2: Basically, our introduction just list some seasonality indexes, which indicate qualitatively the degree of rainfall seasonality in a given precipitation time-series. To assess rainfall seasonality in a quantitative way, among the various existing techniques, the SPI index and Feng's et al. approach appeared to be solid techniques to classify wet and dry months as well as to retrieve precious information on the statistical distribution of daily rainfall values.

3. At line 184 the authors state that they "assumed that the duration of the wet season follows a normal distribution...". While I do not doubt that such hypothesis may be a feasible one, I would expect some kind of validation or testing of it through observed data.
   REPLY-3: We strongly agree with this comment. Accordingly, we will do the Lilliefors test for normality (in MATLAB environment) and re-write the sentence. Thanks for pointing this out.

4. The stochastic Poisson point process with exponential distribution of pulses that is finally used for rainfall generations, I believe could be referenced to classical papers like Rodriguez-Iturbe, I. et al (Journal of Geophysical Research, 1987) and /or Eagleson (WRR, 1972), may be also of interest a more recent application by Veneziano and Iacobellis (WRR, 2002) on Italian datasets, among many others. The use of seasonal parameterization on a stochastic rainfall generator is also a matter of interest.

REPLY-4: We agree and added the three mentioned citations accordingly.

Rodríguez-Iturbe, I., B. Febres de Power, J.B. Valdés. 1987. Rectangular pulses point process models for rainfall: Analysis of empirical data. Journal of Geophysical Research, https://doi.org/10.1029/JD092iD08p09645.

Veneziano, D., V. Iacobellis. 2002. Multiscaling pulse representation of temporal rainfall. Water Resources Research, 38, 1138, 10.1029/2001WR000522

Eagleson, P. S. 1972. Dynamics of flood frequency. Water Resour. Res.,8, 878–898.

5. I believe that also conclusions should be reinforced. First by better depicting which practical use the methodology could be exploited for and, second, by deepening the discussion about the characterization of rainfall seasonality and its anomalies, according to different approaches, which was mentioned as a goal of the study.

REPLY-5: The target of our study was to evaluate the sensitivity of water balance components to rainfall seasonality. This study highlights the importance of accounting for rainfall seasonality in certain regions and under certain conditions, especially if one is interested in building scenario-based projections by using reliable (calibrated/validated) numerical models. We presented results of output fluxes within a probabilistic framework in order to quantify expected and exceptional rainfall seasons in a quantitative reliable procedure. We will give particular attention to this reviewer's suggestion to improve the revised version.

---

## Referee Comment (RC3) · Anonymous Referee #3 · 18 Feb 2020

Review of "Assessing the impact of rainfall seasonality anomalies on catchment-scale water balance components" by Paolo Nasta et al. for HESS

The main research question of this study, as presented by the authors in Line 64, is "What is the impact of rainfall seasonality anomalies on annual-average (or seasonal-average) water supply, and what happens if the Alento River catchment (ARC) will experience several consecutive years of lower-than-expected rainfall events?" The authors use SWAT (Soil Water Assessment Tool) to assess the changes in the different catchment water fluxes in response to changes in rainfall seasonality, using ARC as a study site. The changes in rainfall seasonality is simulated through two different approaches: (i) a "static" approach based on the SPI (Standard Precipitaiton Index) and (ii) a "dynamic" approach by decomposing seasonality into a magnitude, timing, and

duration components following Feng et al. 2013. While simulating the changes in rainfall seasonality via a Monte-Carlo approach, the length of the seasons are set across multiple years but varied across the 3 case scenarios ("reference," "dry," and "wet") for the "static" approach, whereas for the "dynamic" approach, the duration of the wet season in each year is randomly drawn from a normal distribution (line 220 – 222).

To me, the set of main questions is at once too broad ("the effect of rainfall seasonality on the annual catchment water yield") and too specific (effects on one catchment, ARC). The presentation is overall loose and acutely needs focusing. By this I mean that it's not clear to me what conclusions to be drawn from this study other than "by changing rainfall seasonality under scenario X, we simulated a reduction in water yield at this Mediterranean catchment by Y amount," which does not give much scientific insights into how this particular Mediterranean catchment might function (in response to the second part of the main question), nor how the results may be able to be generalized to other Mediterranean catchments around the world (in response to the first part of the main question). Perhaps this is just an issue of having to refine the main question a little more. At one point the authors also state "the goal of this study is to characterize the rainfall seasonality and its anomalies by using two approaches (Line 84)" – to what end? Not only do I find this goal to be a little aimless, but it's also not clear to me how this would help advance the overall research question stated earlier. I understand that this relates to the methodology through which the main questions were interrogated, but why two different approaches? And what did the authors learn from adopting the two different approaches?

The authors claim that the questions of how the catchment water balance plays out in Mediterranean question remains largely unaddressed ("As far as we are aware, there is still a lack of knowledge about the effects of possible changes in rainfall seasonality on the water balance of a catchment subject to a Mediterranean climate, and the analyses presented in this paper aims primarily at contributing to fill this gap." (Lines 84 – 86)) I find this statement to be surprising and again, vague and unrefined, since there is

already a large body of work that already attempts to address this question in one fashion or another, via theoretical and empirical approaches, that remains uncited:

- Potter et al. 2005 "Effects of rainfall seasonality and soil moisture capacity on mean annual water balance for Australian catchments" WRR.

- Hickel and Zhang 2006. "Estimating the impact of rainfall seasonality on mean annual water balance using a top-down approach" JoH.

- Viola et al. 2008 "Transient soil-moisture dynamics and climate change in Mediterranean ecosystems" WRR.

- Gentine et al. 2012 "Interdependence of climate, soil, and vegetation as constrained by the Budyko curve." GRL

- Andersen et al. 2012 "Assessing regional evapotranspiration and water balance across a Mediterranean montane climate gradient." AFM

- Williams et al. 2012 "Climate and vegetation controls on the surface water balance: Synthesis of evapotranspiration measured across a global network of flux towers" WRR

- Feng et al. 2015 "Stochastic soil water balance under seasonal climates" PRSA

- Viola et al. 2019 "Impacts of hydrological changes on annual runoff distribution in seasonally dry basins" WRM

The authors do not make an attempt to contextualize the results of their work against a larger set of studies on water balance in seasonal and Mediterranean climates, and I find this disappointing. My goal in listing these references is not to encourage the authors to simply cite them, but also to use them (amongst others that I have certainly missed) as a starting point to actually pinpoint where the existing knowledge gaps are, and articulate clearly how, using the current approach, they are able to fill them. For example, the fact that we need to account for climate seasonality and non-stationarity when considering annual water balances, to me, does NOT constitute a knowledge

gap – this has been the conclusion of many previous papers.

Other comments:

Line 47: "The amount of rainfall in each season can be suitably decomposed and simulated considering the following three main components." It's not clear to me how this statement fits in with the rest of the introduction. Why is intra-annual variability discussed at this point, when the focus of the study is on inter-annual variability of seasonality? I suggest the authors move this into the method section when discussing the Monte Carlo simulations for daily rainfall. Also, the representation of rainfall via a stochastic Poisson process (which this set of criteria is describing) should be associated with more foundational studies than those of Van Loon et al. 2014 and Feng et al. 2013 – this was introduced first by Rodriguez-Iturbe et al. 1987 "Some models for rainfall based on stochastic point processes" in PRSA and more widely disseminated in Rodriguez-Iturbe et al. 1999, PRSA.

The presentation of Budyko's curve as a conceptual and unifying framework is commendable, but it that it is too rushed. This may be a widely used concept in hydrological sciences, but it does not make a first appearance until the results section (starting on line 367!!) and need to be motivated better in the introduction and methods section.

Additionally, description for each of the "static" scenarios ("reference" "dry" and "wet") also only makes first appearance in the results section (lines 265-270) and need to be moved to the methods section.

SWAT model calibration has not been adequately described. While the performance is shown to be good at the monthly scale (line 141), there could still be compensating model parameters. It would be helpful to see a table of calibrated values for the list of model parameters in lines 137 – 141.
* * *

---

## Author Comment (AC3) · 24 Feb 2020

**Reply to Referee #3**

The main research question of this study, as presented by the authors in Line 64, is "What is the impact of rainfall seasonality anomalies on annual-average (or seasonal-average) water supply, and what happens if the Alento River catchment (ARC) will experience several consecutive years of lower-than-expected rainfall events?" The authors use SWAT (Soil Water Assessment Tool) to assess the changes in the different catchment water fluxes in response to changes in rainfall seasonality, using ARC as a study site. The changes in rainfall seasonality is simulated through two different approaches: (i) a "static" approach based on the SPI (Standard Precipitaiton Index) and (ii) a "dynamic" approach by decomposing seasonality into a magnitude, timing, and duration components following Feng et al. 2013. While simulating the changes in rainfall seasonality via a Monte-Carlo approach, the length of the seasons are set across multiple years but varied across the 3 case scenarios ("reference," "dry," and "wet") for the "static" approach, whereas for the "dynamic" approach, the duration of the wet season in each year is randomly drawn from a normal distribution (line 220 – 222).

**GENERAL REPLY:** We thank this reviewer for her/his comments and suggestions. In the following sections we have tried to provide a few preliminary replies to clarify her/his major concerns.

To me, the set of main questions is at once too broad ("the effect of rainfall seasonality on the annual catchment water yield") and too specific (effects on one catchment, ARC). The presentation is overall loose and acutely needs focusing. By this I mean that it's not clear to me what conclusions to be drawn from this study other than "by changing rainfall seasonality under scenario X, we simulated a reduction in water yield at this Mediterranean catchment by Y amount," which does not give much scientific insights into how this particular Mediterranean catchment might function (in response to the second part of the main question), nor how the results may be able to be generalized to other Mediterranean catchments around the world (in response to the first part of the main question). Perhaps this is just an issue of having to refine the main question a little more. At one point the authors also state "the goal of this study is to characterize the rainfall seasonality and its anomalies by using two approaches (Line 84)" – to what end? Not only do I find this goal to be a little aimless, but it's also not clear to me how this would help advance the overall research question stated earlier. I understand that this relates to the methodology through which the main questions were interrogated, but why two different approaches? And what did the authors learn from adopting the two different approaches?

**REPLY:** Honestly, the first part of this reviewer's comment is not completely clear to us. Firstly, "rainfall seasonality" represents a clear and specific change in the input forcing, whereas "water yield" is a clear and specific output response of a catchment. On the other hand, almost all of the papers we read in the literature refer to a general problem or concern that then is investigated in one specific area where a good amount of quality data is available to elucidate somehow the question at hand. Moreover, especially in recent years, it is a desire to be able to compare outcomes from different sites, an exercise made difficult since only in very few cases the experimental sites are instrumented in similar ways. One eventually tries to get the most from the own site and hopes that these outcomes can be exported to similar sites.

Whereas we do agree with this reviewer that the main research question we pose in this paper should be refined somehow and better worded, an issue relevant to the Mediterranean rainfall seasonality but that does not seem to be well explored yet, at least as we are aware, using the SPI approach is the following: What happens to the water budget if the transition season becomes dry or wet? The "dynamic" approach, instead, identifies two seasons and sets two parameters that characterize the wet season, namely the centroid and spread. The spread of the wet season varies from year to year (inter-annual variability). Therefore, we posed a similar question: what happens to the water budget if the spread of the wet season becomes small (short duration of the wet season, meaning drought) or large (long duration of the wet season)?

By exploiting a long-term rainfall time series, an element of novelty of this manuscript is to assess the impact of wet season duration on the water budget in a river catchment having the UARC features. However, a longer-than-average duration of the wet season does not "always" imply a wetter-thanaverage mean annual rainfall. We do have to take into account also for rainfall magnitude of the wet season. The strategy is to analyze rainfall data and properly characterize the duration and magnitude of rainfall seasons through a Monte-Carlo approach since we want to obtain water budget results within a probabilistic framework.

The authors claim that the questions of how the catchment water balance plays out in Mediterranean question remains largely unaddressed ("As far as we are aware, there is still a lack of knowledge about the effects of possible changes in rainfall seasonality on the water balance of a catchment subject to a Mediterranean climate, and the analyses presented in this paper aims primarily at contributing to fill this gap." (Lines 84 – 86) I find this statement to be surprising and again, vague and unrefined, since there is already a large body of work that already attempts to address this question in one fashion or another, via theoretical and empirical approaches, that remains uncited:

– Potter et al. 2005 "Effects of rainfall seasonality and soil moisture capacity on mean annual water balance for Australian catchments" WRR.
– Hickel and Zhang 2006. "Estimating the impact of rainfall seasonality on mean annual water balance using a top-down approach" JoH.
– Viola et al. 2008 "Transient soil-moisture dynamics and climate change in Mediterranean ecosystems" WRR.
– Gentine et al. 2012 "Interdependence of climate, soil, and vegetation as constrained by the Budyko curve." GRL
– Andersen et al. 2012 "Assessing regional evapotranspiration and water balance across a Mediterranean montane climate gradient." AFM
– Williams et al. 2012 "Climate and vegetation controls on the surface water balance: Synthesis of evapotranspiration measured across a global network of flux towers" WRR
– Feng et al. 2015 "Stochastic soil water balance under seasonal climates" PRSA
– Viola et al. 2019 "Impacts of hydrological changes on annual runoff distribution in seasonally dry basins" WRM

The authors do not make an attempt to contextualize the results of their work against a larger set of studies on water balance in seasonal and Mediterranean climates, and I find this disappointing. My goal in listing these references is not to encourage the authors to simply cite them, but also to use them (amongst others that I have certainly missed) as a starting point to actually pinpoint where the existing knowledge gaps are, and articulate clearly how, using the current approach, they are able to fill them. For example, the fact that we need to account for climate seasonality and non-stationarity when considering annual water balances, to me, does NOT constitute a knowledge gap – this has been the conclusion of many previous papers.

**REPLY:** We are a bit puzzled over that comment. Actually, in the original manuscript we do have cited Potter et al. (2005) (see line 73) and Williams et al. (2012) (see line 393). Other than that, we have cited the papers related to the studies presented by Viola et al. (2019) (see the citations of Viola et al., 2017; Caracciolo et al., 2017 at line 369). Viola et al. (2008) focused on seasonal soil moisture dynamics impacting on plant water stress by using a zero-dimensional bucket-filling model, while ignoring the topographical effect on the lateral distribution, and where the authors identify two seasons and set rainfall parameters arising from a Poisson process. The paper by Anderson et al. (2012) seems a bit on the boundary of the topic of rainfall seasonality. The remaining suggested citations are based on the Budyko approach, but do not focus on the assessment of rainfall seasonality.

Therefore, we aware of the state-of-the-art in the literature and here confirm that, actually, only a few studies (such that of Viola et al, 2008) have dealt in the past with rainfall seasonality issues. Only recently we are witnessing an increase in the number of studies dealing with that topic, and our submission is also heading in this direction. Unless the few previous studies (as for example the paper by Viola et al., 2008), our study proposes a new approach for assessing the impact of observed rainfall data on a water budget. In doing so, we generate new random daily rainfall data as input in a hydrological model (such as SWAT) under a Mediterranean climate. It is therefore fundamental to group rainfall seasons adequately in order to

properly calculate the statistical parameters belonging to a Poisson process. Even when the user has a short-term rainfall data set.

Other comments:

Line 47: "The amount of rainfall in each season can be suitably decomposed and simulated considering the following three main components." It's not clear to me how this statement fits in with the rest of the introduction. Why is intra-annual variability discussed at this point, when the focus of the study is on inter-annual variability of seasonality? I suggest the authors move this into the method section when discussing the Monte Carlo simulations for daily rainfall. Also, the representation of rainfall via a stochastic Poisson process (which this set of criteria is describing) should be associated with more foundational studies than those of Van Loon et al. 2014 and Feng et al. 2013 – this was introduced first by Rodriguez-Iturbe et al. 1987 "Some models for rainfall based on stochastic point processes" in PRSA and more widely disseminated in Rodriguez-Iturbe et al. 1999, PRSA.

**REPLY:** The parameters describing the intra-annual variability of rainfall identify timing, duration, and magnitude of the rainfall seasons (intra-annual variability) that nevertheless change with time (inter-annual variability). We agree with this comment about the seminal paper by Rodriguez-Iturbe et al. (1987), but we did not cite it since it is actually embedded in the papers by van Loon et al. (2014) and Feng et al. (2013).

The presentation of Budyko's curve as a conceptual and unifying framework is commendable, but it that it is too rushed. This may be a widely used concept in hydrological sciences, but it does not make a first appearance until the results section (starting on line 367!!) and need to be motivated better in the introduction and methods section.

**REPLY:** This is a good point and we thank this reviewer for that. Honestly, we should admit that presenting our outcomes even within Budyko's framework is something that was discussed among us only shortly before submitting the manuscript to HESS-D. The revised version of our study will definitely keep this reviewer's suggestion.

Additionally, description for each of the "static" scenarios ("reference" "dry" and "wet") also only makes first appearance in the results section (lines 265-270) and need to be moved to the methods section.

**REPLY:** We agree on that point that helps improve the readability of our paper. A short description will be included in sub-section 4.1 ("Static approach based on the SPI drought index").

SWAT model calibration has not been adequately described. While the performance is shown to be good at the monthly scale (line 141), there could still be compensating model parameters. It would be helpful to see a table of calibrated values for the list of model parameters in lines 137 – 141.

**REPLY:** This concern was raised also by Reviewer#1. Below we report our reply:

Nasta et al. (2017 STotEnv) calibrated a few model parameters by comparing measured and simulated monthly water yield values recorded at the dam. Numerical simulations were run at daily time step (the only time step allowed in SWAT). In this study, we followed the same criterion, we run numerical simulations at daily time step (rainfall was randomly generated at daily time step) and aggregated the output fluxes at monthly time resolution. We are aware that calibrating at monthly time-scale might lead to a potential misfit between measured and simulated values at daily time-scale (e.g. Adla et al., 2019). However, our analysis is based on the monthly aggregation of fluxes and we analyzed seasonal patterns of monthly aggregates. This is an important point that requires to be clarified in the revised manuscript (Adla, S., S. Tripathi, M. Disse, 2019. Can we calibrate a daily time-step hydrological model using monthly time-step discharge data? Water 11, 1750; doi:10.3390/w11091750).

---

## Author Response (AR1)

**Reply to the Handling Editor and Reviewers**

We thank the Editor for handling the review process, and the three anonymous Reviewers for evaluating our work. We have implemented all applicable recommendations that improve the quality of the original manuscript. Some preliminary replies to the comments received were posted already during the public discussion step of the journal, and more targeted point-by-point responses are given below. To avoid confusion, all line numbers refer to the revised version of the manuscript (attached to the response letter), where all changes are properly tracked. Finally, please note a native English speaker, with expertise in environmental sciences, has fine-tuned all parts of the revised paper and this reply.

**Reply to the Handling Editor**

Dear Authors, after receiving three reviews, and by reading your answers to the comments and suggestions made by the reviewers, I am convinced that the manuscript can be enhanced and made more readable with a clear research question(s) and final message what your research results bring to the scientific community, at least in the Mediterranean area.

Please, prepare a revised manuscript at your earliest convenience by incorporating into it your answers from the discussion phase.

The revised manuscript will be reviewed again by at least one reviewer and myself.

Sincerely Yours,

MatjažMikoš

Handling Editor

**REPLY:** We thank the Editor for handling the review process of our original submission and for providing useful recommendations on how to improve the manuscript. In the following, we provide our final replies to the three reviewers who evaluated our original paper during the public discussion step of the journal. We tried to thoroughly reformulate the last part of the Introduction (lines 64-100) to clarify the research question, the goal of our study, the novelty proposed by this study and the conclusions. We followed almost all comments and suggestions received from the three reviewers, especially by improving the description of the methods and clarity in reporting results and comments.

**General comments from the authors on reviewers' reports**

Three reviewers evaluated our original manuscript, making interesting comments but also raising some concerns. A few concerns refer to general matters, whereas the rest are mostly linked to their personal views on the topic of our study. While there is a consensus on the quality with which we presented our investigation, the opinions were somewhat more critical on the scientific significance and how we discussed our results.

In the light of the comments and recommendations received, we have revised the introduction and better focused (we do hope) the main research question that guided our present study. We have put considerable effort into the revised manuscript to clarify those parts that might give rise to strong criticisms. In the event of disagreement with some reviewer's criticism, we shall give adequate reasons for our position.

We also changed the title. The new one is: "Assessing the impact of seasonal rainfall anomalies on catchment-scale water balance components".

**Reply to Referee #1**

COMMENT 1.1. General comments: In the submitted paper, authors investigate the impact of the rainfall seasonality anomalies on the catchment water balance components. For this purpose, a catchment in the southern Italy is selected and SWAT model is applied in order to carry out the investigation. Two different approaches are used in order to define rainfall scenarios. First approach is based on the standardized precipitation index and second one is based on the duration of the wet season as proposed by Feng et al. (2013). The topic is potentially interesting for the society and HESS readers.

**REPLY-1.1.** We thank this reviewer for her/his positive comments and appreciation of the potential interest in our work.

However, two main shortcomings of the paper from my perspective that should be improved are:

COMMENT 1.2. Firstly, the main focus of the paper is to investigate what is the impact of different rainfall scenarios on the water yield, actual evapotranspiration and groundwater recharge. Thus, for different scenarios changes in these variables are analyzed with respect to reference case. Model calibration is just briefly described and reference to more detailed description is given (Nesta et al., 2017). It seems that model was calibrated using monthly data (?). However, P6, L124 states that daily time step of the SWAT model was used. I think that model should also be calibrated using daily data if authors want to use this time step. Otherwise, I would suggest to aggregate daily rainfall data into monthly and re-run the model with monthly time step (if this is possible or perhaps use a different model). An alternative is, to calibrate the model using daily data if there is a discharge gauging station available near the catchment outlet.

**REPLY-1.2.** Nasta et al. (2017 STotEnv) calibrated nine model parameters by comparing measured and simulated monthly water yields recorded at the dam. Numerical simulations were run at the daily time step (the only time step allowed in SWAT). In this study, we followed the same criterion: we ran numerical simulations at the daily time step (rainfall was randomly generated at the daily time step) and aggregated the output fluxes at a monthly time resolution. We are aware that calibrating at the monthly time-scale might lead to a potential misfit between measured and simulated values at a daily time-scale (e.g. Adla et al., 2019, Water). However, our analysis is based on the monthly aggregation of fluxes and we analyzed seasonal patterns of monthly aggregates. In the light of the above comment, we added a new part at lines 182-188 to clarify this important point and why this misfit should not be viewed as relevant to our analysis. The reference to the paper by Adla et al. (2019) is also added.

Adla, S., S. Tripathi, M. Disse, 2019. Can we calibrate a daily time-step hydrological model using monthly time-step discharge data? Water 11, 1750; doi:10.3390/w11091750.

COMMENT 1.3. Secondly, when using different scenarios, authors only modified rainfall characteristics, what about air temperature? It is true that in some cases the dependence between these two variables can be low or even none existing. However, is some other cases, some dependence could exist. For example, higher average annual temperature could lead to lower annual rainfall and vice-versa. Or higher daily temperature in summer could cause higher rainfall amounts due to more extreme thunderstorm. Did authors check the relationship for this specific catchment? Moreover, I think that air temperature variability should be included in this kind of investigations. Even if there is no clear relationship with rainfall.

**REPLY-1.3.** We fully agree with this comment, which in our opinion is timely. Indeed, the feedback of temperature on precipitation is a widely recognized cause for the increasing frequency of storms and heavy rainfall events under climate change. Unfortunately, we do not have easy access to a database of long time series of observed temperatures to reliably evaluate this linkage in our case study. However, one should note that this kind of information cannot be restricted only to the surrounding air temperatures, but should include the change in the sea surface temperature and air moisture to correctly frame the analysis. This is certainly an issue that deserves an in-depth investigation, but it goes beyond the scope of our study. We focused only on the long-term (almost one century) daily time series of rainfall, yielding interesting outcomes about the sensitivity of catchment hydrological response to seasonal rainfall patterns alone. Moreover, for the same reason (lack of comprehensive spatial and temporal sets of temperature data), we were unable to capture any significant (increasing) trend and assumed the temperature is stationary in time and so is evapotranspiration (although, of course we reproduce the seasonal cycle). Our analysis addressed only the impact exerted by rainfall seasonality on the major water balance components. The availability of adequate datasets should highlight whether the observed increases in air temperatures impact the daily precipitations, or mostly only the sub-daily and sub-hourly rainfall values. In any case, we took care of temperature variability (and temporal variability of all weather data) because daily potential evapotranspiration data were calculated by using random values of weather data drawn from their normal distribution in each month of the year. We reformulated this part by clarifying sentences (lines 167-172)

Specific comments:
I would suggest to add a figure showing the location of the catchment with stations used.
**REPLY-1.4.** We added the new Fig. 1 and have accordingly changed the figure numbering throughout the revised manuscript.

P6, L130: Please better explain what is meant by the term boundary forcings.
**REPLY-1.5.** With the term "boundary forcings", we mean the (input) water fluxes (rainfall and potential evapotranspiration) set as upper boundary conditions to the flow domain.
To avoid misunderstanding we rephrased all occurrences of "boundary forcings" with "rainfall and potential evapotranspiration forcings". We modified also y-axis title in Fig. 10 as well.

P7, L142-144: Why did you used only 3 years for simulation and why 2-years warm-up period? How does this selection impact on the results? Moreover, does initial state of the catchment also has impact on the results (i.e. using different initial values of model variables)?
**REPLY-1.6.** We decided to run three years in each scenario and neglect the model simulations of the first two years to annihilate the impact of initial soil moisture values set in the soil domain. We point out that the soil moisture content at the initial day of year 1 is set at the value of "field capacity" (which can be already considered a realistic situation in winter under Mediterranean climate). Moreover, we have considered the third year of model simulation. We repeat this exercise 10,000 times so as to frame the output fluxes within a probabilistic framework. We added this clarification in the text (see lines 189-196)

P7, L146-149: The data from other station will be used for analyses at monthly time scale but the model will run with daily time step and daily reference evapotranspiration will be calculated? Perhaps you could rephrase this sentence.
**REPLY-1.7.** We agree and we reformulate this (unclear) sentence. We have the daily weather values and we use the descriptive statistics of daily values in each month of the year to generate new random daily values of evapotranspiration in each month. See also reply1.3. Sentences in lines 196-200 were rephrased.

P7, L149: Here reference evapotranspiration is mentioned but in next sections, you only mention potential and actual evapotranspiration. Why was reference evapotranspiration used?
**REPLY-1.8.** SWAT uses weather data to estimate potential evapotranspiration ($ET_p$). We replaced the word "reference" with the word "potential" at line 201. Thanks for pointing it out.

P9, section 4.3: If I understand correctly exponential distribution was selected only based on the graphical comparison shown in Figure 2 and Figure 3? If this is the case, I would suggest to additionally apply a suitable statistical test.
**REPLY-1.9.** The stochastic Poisson point process of daily rainfall occurrences was assumed to represent daily rainfall evolution for its easy reproducibility (Lines 252-254). In a preliminary analysis, we tested it and compared it with the best parent distribution, namely the Generalized Pareto Distribution. In this case, we observed a fair agreement between the two models for representing the daily rainfall evolution recorded at the Gioi Cilento weather station, and concluded that the simple-to-use exponential model is suitable (Lines 267-269).
The stochastic Poisson point process is widely used for its simplicity and parsimony as pointed out by Reviewer#2 (we list below the three references reported in Reply 2.4 to Reviewer #2).

Rodríguez-Iturbe, I., B. Febres de Power, J.B. Valdés. 1987. Rectangular pulses point process models for rainfall: Analysis of empirical data. Journal of Geophysical Research, https://doi.org/10.1029/JD092iD08p09645
Veneziano, D., V. Iacobellis. 2002. Multiscaling pulse representation of temporal rainfall. Water Resources Research, 38, 1138, 10.1029/2001WR000522
Eagleson, P. S. 1972. Dynamics of flood frequency. Water Resour. Res., 8, 878–898.

P10, L230-231: I do not understand this sentence, if you split the data, how can you then have a drying trend? Only for the second 45 years?

**REPLY-1.10.** The frequency distribution of SPI-6 values in the first 45 years is wetter than the one pertaining to the last (more recent) 45 years. This is now clarified at Lines 306-312.

P13, L285 and L294: A statistical test is mentioned here but no information about null and alternative hypothesis is given. Moreover, authors should rephrase these sentences. In statistical hypothesis testing the null hypothesis can be either rejected in favor of the alternative hypothesis or cannot be rejected (with the chosen significance level). Moreover, all the methods used should probably be mentioned and described in the methodology section (and not results and discussion).

**REPLY-1.11.** The Mann-Kendal test has become a standard test to search for a possible trend in a time series and is widely applied in the literature. Therefore, after a discussion among the co-authors, we decided not to describe the Mann-Kendal test in detail and just mentioned the reference.

Sections 5.3 and 5.4 and conclusions: The main results of the paper are somehow expected: dry scenario leads to less runoff, groundwater recharge and also less actual evapotranspiration (compared to reference scenario). On the other hand, wet scenario leads to more runoff, groundwater recharge and actual evapotranspiration (compared to reference scenario). Moreover, different rainfall simulation methods yield different results. The actual relationship among variables mostly depends on the rainfall characteristics, especially if variability in air temperature is not considered. Can the authors perhaps somehow enhance the take home message of this paper?

**REPLY-1.12.** The target of our study is to evaluate the sensitivity of some water balance components to seasonal rainfall anomalies (potential temperature effects are not considered here, partly because of the lack of suitable datasets). We thoroughly reformulated the conclusion section by highlighting the take-home message of this paper. We recalled the main research question that we posed in the Introduction: "What is the impact of seasonal rainfall anomalies on annual-average (or seasonal-average) water supply in UARC?"

We briefly present the following steps to answer the aforementioned question: 1) we needed to build robust scenarios based on well-posed hydrological model (SWAT) by presenting results within a probabilistic framework; 2) to do that, we need to analyze the long-term historical rainfall time series and identify rainfall seasons; 3) evaluate the best statistical distribution of rainfall daily values (Poisson model) in each season; 4) propose two approaches to detect seasonal rainfall anomalies and stress pros and cons.

Moreover, the assumption of the steady-state condition inherent in the Budyko approach is questioned. The stationarity/non-stationarity dilemma in hydrological processes is still a matter of an open debate in the scientific community (Milly et al., 2008; Montanari and Koutsoyiannis, 2014).

Milly, P. C. D., J. Betancourt, M. Falkenmark, R. M. Hirsch, Z. W. Kundzewicz, D. P. Lettenmaier, and R. J. Stouffer. 2008. Stationarity is dead: Whither water management?Science,319:573–574.

Montanari, A., and D. Koutsoyiannis. 2014. Modeling and mitigating natural hazards: Stationarity is immortal!Water Resour. Res.,50:9748–9756.

**Reply to Referee #2**

The paper deals with the assessment of water balance components (i.e. water yield, evapotranspiration, groundwater recharge, etc.) and relative deficit in case of climatic anomalies related to seasonality in a Mediterranean basin. This is done by parameterizing a rainfall generator model according to two different schematic representation of seasonality (called "static" and "dynamic"), and using synthetic rainfall series as input to the SWAT hydrological model. While shifts and changes in seasonal patterns have been addressed by many researchers as key factors in analyzing the hydrological impact of climatic fluctuations, the consequent issue of how these phenomena may impact the regulation of artificial reservoirs, designed for annual or multiyear storage purpose, deserves attention.

**GENERAL REPLY:** We thank this reviewer for her/his comments and suggestions.

The paper is in general well sounded and relevant although it could be improved in my opinion, accounting for the following suggestions. The paper is compound by two main issues:

2.1. The first one is referred to the analysis of the climatic forcing and the parameterization of the rainfall model; the second one is related to the use of SWAT model to obtain different components of water balance. A stronger emphasis is given to the first one, which is also performed by comparing different methods, while the second one is much less discussed. Also, the overall paper goal could be better assessed and the methodology more detailed in the introduction. To make an example, the sentence "The goal of the study is to characterize the rainfall seasonality and its anomalies by using two approaches." (line 81) is in my opinion somehow misleading with respect to the overall paper objectives and developments.

**REPLY-2.1.** We agree with this concern that was also raised by Ref.#3 (see our Reply 3.1). Therefore, we have completely reformulated the last part of the Introduction (Lines 69-117) and overhauled the conclusions (Lines 523-583) to make the paper more effective and clarify our goals and take-home messages (see also Reply 1.12 to Ref.#1).

2.2. Dealing with issue #1, i.e. seasonality assessment, in the introduction the PCI and SI methods are indicated as most popular approaches. Nevertheless, the authors do not use them but rather prefer an SPI based analysis and the procedure proposed by Feng et al (2013). A better acknowledgement could be provided about the reasons of such choices, and the comparisons between the performances of different methods.

**REPLY-2.2.** Basically, our introduction lists some seasonality indexes, which indicate qualitatively the degree of rainfall seasonality in a given precipitation time-series. To assess rainfall seasonality quantitatively, among the various existing techniques, the SPI index and the Feng et al. approach appeared to be sound techniques to classify wet and dry months as well as to retrieve precious information on the statistical distribution of daily rainfall values.
We added the following new sentence in lines 81-96: "Nonetheless, while PCI and SI are useful indexes to classify rainfall seasonality and the degree of concentration of rainfall within the year, their implementation in a Monte Carlo framework is not straightforward. Therefore, we opted to characterize rainfall seasonality and its anomalies by using the two approaches described as follows. A first approach, which is hereafter referred to as the static approach, is based on the analysis of the standardized precipitation index (SPI) to define the duration of a wet season (4 months), a dry season (4 months) and a transition season (2 months from dry to wet phase plus 2 months from wet to dry phase) in UARC. In this approach, the drought anomaly is rigidly built with the artifact of extending the duration of the dry season to eight months by removing the transition season. The same criterion applies to a prolonged duration of the rainy season. The second approach, instead, exploits the seasonality characterization proposed by Feng et al. (2013) and can be viewed as a dynamic approach since the duration of the rainy season is time-variant (inter-annual variability) and can be stochastically generated with random duration values drawn from their statistical distribution. This second approach investigates what happens to the water budget if the duration of the rainy season becomes shorter-than-normal (i.e. rainfall scarcity) or longer-than-normal (i.e. rainfall excess). As far as we are aware, there is still a lack of knowledge about the effects of possible changes in rainfall seasonality on the water balance of a catchment subject to a Mediterranean climate, and the analyses presented in this paper aim primarily to contribute to fill this gap.".

2.3. At line 184 the authors state that they "assumed that the duration of the wet season follows a normal distribution...". While I do not doubt that such hypothesis may be a feasible one, I would expect some kind of validation or testing of it through observed data.

**REPLY-2.3.** We strongly agree with this comment. Actually, we applied the Lilliefors test for normality in Section 5.2 (lines 382-391), and also anticipated this result in section 4.2 (lines 238-243).

2.4. The stochastic Poisson point process with exponential distribution of pulses that is finally used for rainfall generations, I believe could be referenced to classical papers like Rodriguez-Iturbe, I. et al (Journal of Geophysical Research, 1987) and /or Eagleson (WRR, 1972), may be also of interest a more recent application by Veneziano and Iacobellis (WRR, 2002) on Italian datasets, among many others. The use of seasonal parameterization on a stochastic rainfall generator is also a matter of interest.

**REPLY-2.4.** We agree and added the three mentioned citations accordingly (lines 254-255).

Rodríguez-Iturbe, I., B. Febres de Power, J.B. Valdés. 1987. Rectangular pulses point process models for rainfall: Analysis of empirical data. Journal of Geophysical Research, https://doi.org/10.1029/JD092iD08p09645.

Veneziano, D., V. Iacobellis. 2002. Multiscaling pulse representation of temporal rainfall. Water Resources Research, 38, 1138, 10.1029/2001WR000522

Eagleson, P.S. 1972. Dynamics of flood frequency. Water Resour. Res.,8, 878–898.

2.5. I believe that also conclusions should be reinforced. First by better depicting which practical use the methodology could be exploited for and, second, by deepening the discussion about the characterization of rainfall seasonality and its anomalies, according to different approaches, which was mentioned as a goal of the study.

**REPLY-2.5.** We strongly agree with this comment, which was also stated by Ref.#1 (Reply1.12). We therefore report the same reply below.

The target of our study is to evaluate the sensitivity of water balance components to seasonal rainfall anomalies (potential temperature effects are not considered here, partly because of the lack of suitable datasets). We thoroughly reformulated the Conclusions Section by highlighting the take-home message of this paper. We recalled the main research question that we posed in the Introduction: "What is the impact of seasonal rainfall anomalies on annual-average (or seasonal-average) water supply in UARC?"

We briefly present the following steps to answer the aforementioned question: 1) we needed to build robust scenarios based on well-posed hydrological model (SWAT) by presenting results within a probabilistic framework; 2) to do that, we need to analyze the long term historical rainfall time series and identify rainfall seasons; 3) evaluate the best statistical distribution of rainfall daily values (Poisson model) in each season; 4) propose two approaches to detect rainfall seasonality anomalies and stress pros and cons.

Moreover, the assumption of the steady-state condition inherent in the Budyko approach is questioned. The stationarity/non-stationarity dilemma in hydrological processes is still a matter of an open debate in the scientific community (Milly et al., 2008; Montanari and Koutsoyiannis, 2014).

Milly, P. C. D., J. Betancourt, M. Falkenmark, R. M. Hirsch, Z. W. Kundzewicz, D. P. Lettenmaier, and R. J. Stouffer. 2008. Stationarity is dead: Whither water management? Science, 319:573–574.

Montanari, A., and D. Koutsoyiannis. 2014. Modeling and mitigating natural hazards: Stationarity is immortal! Water Resour. Res., 50:9748–9756.

**Reply to Referee #3**

The main research question of this study, as presented by the authors in Line 64, is "What is the impact of rainfall seasonality anomalies on annual-average (or seasonal-average) water supply, and what happens if the Alento River catchment (ARC) will experience several consecutive years of lower-than-expected rainfall events?" The authors use SWAT (Soil Water Assessment Tool) to assess the changes in the different catchment water fluxes in response to changes in rainfall seasonality, using ARC as a study site. The changes in rainfall seasonality is simulated through two different approaches: (i) a "static" approach based on the SPI (Standard Precipitaiton Index) and (ii) a "dynamic" approach by decomposing seasonality into a magnitude, timing, and duration components following Feng et al. 2013. While simulating the changes in rainfall seasonality via a Monte-Carlo approach, the length of the seasons are set across multiple years but varied across the 3 case scenarios ("reference," "dry," and "wet") for the "static" approach, whereas for the "dynamic" approach, the duration of the wet season in each year is randomly drawn from a normal distribution (line 220 – 222).

**GENERAL REPLY:** We thank this reviewer for her/his comments and suggestions.

To me, the set of main questions is at once too broad ("the effect of rainfall seasonality on the annual catchment water yield") and too specific (effects on one catchment, ARC). The presentation is overall loose and acutely needs focusing. By this I mean that it's not clear to me what conclusions to be drawn from this study other than "by changing rainfall seasonality under scenario X, we simulated a reduction in water yield at this Mediterranean catchment by Y amount," which does not give much scientific insights into how this particular Mediterranean catchment might function (in response to the second part of the main question), nor how the results may be able to be generalized to other Mediterranean catchments around the world (in response to the first part of the main question). Perhaps this is just an issue of having to refine the main question a little more. At one point the authors also state "the goal of this study is to characterize the rainfall seasonality and its anomalies by using two approaches (Line 84)" – to what end? Not only do I find this goal to be a little aimless, but it's also not clear to me how this would help advance the overall research question stated earlier. I understand that this relates to the methodology through which the main questions were interrogated, but why two different approaches? And what did the authors learn from adopting the two different approaches?

**REPLY-3.1.** Almost all of the papers we read in the literature refer to a general problem or concern that seasonality is investigated in one specific area where a good amount of quality data is available to elucidate somehow the question at hand. Moreover, especially in recent years, it is desirable to compare outcomes from different sites, an exercise made difficult since only in very few cases are the experimental sites instrumented in similar ways. One eventually tries to get the most from one's own site and hopes that these outcomes can be exported to similar sites.

While we do agree with this reviewer that the main research question we pose in this paper should be refined somehow and better worded, we are confident that the "static" and "dynamic" implementations discussed in the manuscript will contribute giving answer to some timely but still unexplored (at our best knowledge) issues, that are relevant to the Mediterranean rainfall seasonality. Specifically, the "static" approach (based on SPI) addresses the issue *"What happens to the water budget if the transition season becomes dry or wet?"*; while the "dynamic" approach, allowing the wet season to vary from year to year and thus accounting for inter-annual variability, aims to answer the question *"What happens to the water budget if the spread of the wet season becomes smaller-than-average (short duration of the wet season, meaning drought) or larger-than-average (long duration of the wet season)?"*

By exploiting a long-term rainfall time series, an element of novelty of this manuscript is to assess the impact of wet season duration on the water budget in a river catchment with the UARC features. However, a longer-than-average duration of the wet season does not "always" imply a wetter-than-average mean annual rainfall. We do have to take into account also rainfall magnitude of the wet season. The strategy is to analyze rainfall data and properly characterize the duration and magnitude of rainfall seasons through a Monte-Carlo approach since we want to obtain water budget results within a probabilistic framework.

In light of the above comment and also the other two reviewers' comments, we completely changed the last part of the Introduction. Please see lines 69-117 and by following similar concerns raised by Reviewer#1 (Reply 1.12) and Reviewer#2 (Reply 2.5) overhauled the Conclusions (Lines 523-583)

The authors claim that the questions of how the catchment water balance plays out in Mediterranean question remains largely unaddressed ("As far as we are aware, there is still a lack of knowledge about the effects of possible changes in rainfall seasonality on the water balance of a catchment subject to a Mediterranean climate, and the analyses presented in this paper aims primarily at contributing to fill this gap." (Lines 84 – 86) I find this statement to be surprising and again, vague and unrefined, since there is already a large body of work that already attempts to address this question in one fashion or another, via theoretical and empirical approaches, that remains uncited:

- Potter et al. 2005 "Effects of rainfall seasonality and soil moisture capacity on mean annual water balance for Australian catchments" WRR.
- Hickel and Zhang 2006. "Estimating the impact of rainfall seasonality on mean annual water balance using a top-down approach" JoH.
- Viola et al. 2008 "Transient soil-moisture dynamics and climate change in Mediterranean ecosystems" WRR.
- Gentine et al. 2012 "Interdependence of climate, soil, and vegetation as constrained by the Budyko curve." GRL
- Andersen et al. 2012 "Assessing regional evapotranspiration and water balance across a Mediterranean montane climate gradient." AFM
- Williams et al. 2012 "Climate and vegetation controls on the surface water balance: Synthesis of evapotranspiration measured across a global network of flux towers" WRR

- Feng et al. 2015 "Stochastic soil water balance under seasonal climates" PRSA
- Viola et al. 2019 "Impacts of hydrological changes on annual runoff distribution in seasonally dry basins" WRM

The authors do not make an attempt to contextualize the results of their work against a larger set of studies on water balance in seasonal and Mediterranean climates, and I find this disappointing. My goal in listing these references is not to encourage the authors to simply cite them, but also to use them (amongst others that I have certainly missed) as a starting point to actually pinpoint where the existing knowledge gaps are, and articulate clearly how, using the current approach, they are able to fill them. For example, the fact that we need to account for climate seasonality and non-stationarity when considering annual water balances, to me, does NOT constitute a knowledge gap – this has been the conclusion of many previous papers.

**REPLY-3.2.** Actually, in the original manuscript we did cite Potter et al. (2005) (see line 73) and Williams et al. (2012) (see line 393). Other than that, we have cited the papers related to the studies presented by Viola et al. (2019) (see the citations of Viola et al., 2017; Caracciolo et al., 2017 at line 369). Viola et al. (2008) focused on seasonal soil moisture dynamics impacting on plant water stress by using a zero-dimensional bucket-filling model, while ignoring the topographical effect on the lateral distribution, and where the authors identify two seasons and set rainfall parameters arising from a Poisson process. The paper by Anderson et al. (2012) seems a bit on the boundary of the topic of rainfall seasonality. The remaining citations suggested are based on the Budyko approach, but do not focus on the assessment of rainfall seasonality.

Therefore, we are aware of the state of the art in the literature and here confirm that, actually, few studies (such that of Viola et al, 2008) have dealt in the past with rainfall seasonality issues. Only recently have we witnessed an increase in the number of studies dealing with that topic, and our submission is also heading in this direction. Unlike the few previous studies (such as the paper by Viola et al., 2008), our study proposes a new approach for assessing the impact of observed rainfall data on a water budget. In so doing, we generate new random daily rainfall data as input in a hydrological model (such as SWAT) under a Mediterranean climate. It is therefore fundamental to group rainfall seasons adequately to properly calculate the statistical parameters belonging to a Poisson process even when the user has a short-term rainfall data set.

We gave due consideration to this comment (please see Reply-3.1) and changed several parts of the Introduction (lines 69-117)

Other comments:

Line 47: "The amount of rainfall in each season can be suitably decomposed and simulated considering the following three main components." It's not clear to me how this statement fits in with the rest of the introduction. Why is intra-annual variability discussed at this point, when the focus of the study is on inter-annual variability of seasonality? I suggest the authors move this into the method section when discussing the Monte Carlo simulations for daily rainfall. Also, the representation of rainfall via a stochastic Poisson process (which this set of criteria is describing) should be associated with more foundational studies than those of Van Loon et al. 2014 and Feng et al. 2013 – this was introduced first by Rodriguez-Iturbe et al. 1987 "Some models for rainfall based on stochastic point processes" in PRSA and more widely disseminated in Rodriguez-Iturbe et al. 1999, PRSA.

**REPLY-3.3.** The parameters describing the intra-annual variability of rainfall identify the timing, duration, and magnitude of the rainfall seasons (intra-annual variability) that nevertheless change with time (inter-annual variability). We agree with this comment about the seminal paper by Rodriguez-Iturbe et al. (1987), but we did not cite it since it is actually embedded in the papers by van Loon et al. (2014) and Feng et al. (2013).

The presentation of Budyko's curve as a conceptual and unifying framework is commendable, but it that it is too rushed. This may be a widely used concept in hydrological sciences, but it does not make a first appearance until the results section (starting on line 367!!) and need to be motivated better in the introduction and methods section.

**REPLY-3.4.** This is a good point and we thank this reviewer for that. Honestly, we should admit that presenting our outcomes even within Budyko's framework is something that was discussed among us only shortly before submitting the manuscript to HESS-D. In the revised paper Budyko's theory has been moved from Section 5.4 to Section 4.3 (lines 289-300)

Additionally, description for each of the "static" scenarios ("reference" "dry" and "wet") also only makes first appearance in the results section (lines 265-270) and need to be moved to the methods section.
**REPLY-3.5.** In this case, we prefer to keep this description as in the original manuscript, because it is based on the results rather than being an a-priori hypothesis.

SWAT model calibration has not been adequately described. While the performance is shown to be good at the monthly scale (line 141), there could still be compensating model parameters. It would be helpful to see a table of calibrated values for the list of model parameters in lines 137 – 141.
**REPLY-3.6.** This concern was raised also by Reviewer#1. Below we report our reply 1.2:
Nasta et al. (2017 STotEnv) calibrated nine model parameters by comparing measured and simulated monthly water yields recorded at the dam. Numerical simulations were run at the daily time step (the only time step allowed in SWAT). In this study, we followed the same criterion: we ran numerical simulations at the daily time step (rainfall was randomly generated at the daily time step) and aggregated the output fluxes at a monthly time resolution. We are aware that calibrating at the monthly time-scale might lead to a potential misfit between measured and simulated values at a daily time-scale (e.g. Adla et al., 2019, Water). However, our analysis is based on the monthly aggregation of fluxes and we analyzed seasonal patterns of monthly aggregates. In the light of the above comment, we added a new part at lines 182-188 to clarify this important point and why this misfit should not be viewed as relevant to our analysis. The reference to the paper by Adla et al. (2019) is also added.

[revised manuscript text omitted]

---

## Editor Decision (ED1)

[revised manuscript text omitted]

---

## Author Response (AR2)

**Università degli Studi di Napoli Federico II**

**DIPARTIMENTO DI AGRARIA - *DEPARTMENT OF AGRICULTURE**

SEZIONE: INGEGNERIA AGRARIA, FORESTALE E DEI BIOSISTEMI

*DIVISION OF AGRICULTURAL, FOREST AND BIOSYSTEMS ENGINEERING*

Via Università, n. 100  –  80055 Portici (Napoli), ITALY | Tel.: +39 081 2539417

**Hydrology and Earth System Sciences**

**RE: submission of manuscript "Assessing the impact of seasonal rainfall anomalies on catchment-scale water balance components" by Paolo Nasta, Carolina Allocca, Roberto Deidda, Nunzio Romano**

Dear Editor, Prof. Matjaz Mikos,

On behalf of my coauthors, I wish to submit the manuscript including technical corrections reported in the pdf file.

Yours sincerely,

Paolo Nasta

Department of Agriculture, Division of Agricultural, Forest and Biosystems Engineering
University of Napoli Federico II,
Portici (Napoli), Italy
E-mail: paolo.nasta@unina.it
Tel: +39-081-2539417